# PQ-Net: Periodic Quantum Networks for Multivariate Time Series Forecasting

## Abstract

Multivariate time series forecasting (MTSF) requires capturing both periodic structures and cross-channel dependencies from complex temporal signals. To address this challenge, we propose Periodic Quantum Networks (PQ-Net), a quantum–classical hybrid forecasting architecture that integrates a learnable temporal query mechanism for cycle alignment and a channel aggregation module for modeling inter-channel correlations. PQ-Net preserves permutation equivariance across variables while jointly representing frequency-domain and cross-channel information in a principled manner. At the core of PQ-Net lies the Data Reuploading Quantum Circuit (DRQC), whose representational capacity we theoretically analyze. We show that DRQC are mathematically equivalent to truncated Fourier series, enabling natural encoding of periodic patterns, while quantum entanglement provides a means to capture inter-variable dependencies. This interpretation establishes DRQC as a rigorous and interpretable foundation for periodic modeling within PQ-Net. Extensive experiments on twelve real-world datasets demonstrate that PQ-Net consistently achieves state-of-the-art forecasting accuracy over strong classical and quantum baselines, and preliminary hardware results further validate its practicality on real quantum devices.

## 1 Introduction

Multivariate Time Series Forecasting (MTSF) is a fundamental task with broad applications in domains such as traffic management, climate science, and healthcare (Box et al., 2015; Brownlee, 2017; Qiu et al., 2024). A key characteristic of real-world time series is the prevalence of periodic and quasi-periodic patterns, ranging from daily and weekly traffic rhythms to seasonal climate cycles and oscillatory physiological signals (Wang et al., 2024b). Exploiting such regularities can substantially enhance long-horizon forecasting; however, this remains challenging when multiple incommensurate periods overlap, phases drift over time, and observations are corrupted by noise or missing values (Wen et al., 2023; Yan et al., 2021).

In the literature, period-aware models such as CycleNet (Lin et al., 2024) incorporate explicit periodic inductive biases, aligning more closely with underlying cycles. However, they lack a unified mechanism that can simultaneously represent periodic structure while capturing rich cross-variable interactions, both of which are crucial for MTSF. In contrast, attention-based architectures, including iTransformer (Liu et al., 2024) and TimeXer (Wang et al., 2024a), demonstrate strong modeling capacity through adaptive dependency learning, yet their heavy reliance on self-attention makes them vulnerable to noise, particularly in high-dimensional settings.

These observations highlight the need for a modeling paradigm that can jointly encode periodic structures and capture complex inter-variable dependencies, while remaining robust to noise. To this end, we introduce a quantum–classical hybrid perspective. We introduce Periodic Quantum Networks (PQ-Net), a modular forecaster for MTSF that unifies periodic-structure modeling with cross-variable interaction learning. At its core lies the Data-Reuploading Quantum Circuits (DRQC) module, which directly encodes periodic structures into a quantum–classical latent space while leveraging quantum entanglement to capture cross-variable dependencies. Unlike purely classical architectures, DRQC naturally supports phase-consistent frequency composition and admits a rigorous Fourier-series interpretation (Schuld et al., 2020; Zhao et al., 2024), thereby providing a principled and interpretable foundation for periodic modeling. Moreover, DRQC can faithfully re-

construct both oscillatory backbones and sharp spike phases, while avoiding overemphasis on spurious high-frequency ripples or phase drift. To further substantiate these advantages, we conduct dedicated experiments in the experimental section to evaluate its periodic modeling capacity.

In addition, PQ-Net incorporates Instance Normalization (IN) (Kim et al., 2021) to stabilize optimization and enhance cross-channel comparability; Learnable Periodic Vectors (LPV), inspired by CycleNet (Lin et al., 2024), to provide phase-aligned periodic priors; and stackable DRQC blocks that capture spectral structure while modeling variable entanglement, followed by an MLP projection head for prediction.

The main contributions of this study are summarized as follows:

- We propose the Periodic Quantum Networks (PQ-Net), a quantum–classical hybrid framework that jointly integrates explicit periodic-structure modeling with expressive cross-variable dependency learning for multivariate time series forecasting.
- We provide a theoretical analysis showing that the Data Re-uploading Quantum Circuits (DRQC) can encode periodic patterns through its equivalence to Fourier-series modeling and capture inter-variable dependencies via quantum entanglement, a conclusion that is further corroborated by empirical evidence from periodic modeling experiments.
- We conduct extensive experiments on 12 real-world multivariate datasets, demonstrating that PQ-Net substantially outperforms quantum–classical hybrid baselines, periodicity-aware models, and attention-based benchmarks in overall performance. Moreover, preliminary evaluations on real quantum hardware validate the practicality of our approach.

## 2 RELATED WORK

### 2.1 MULTIVARIATE TIME-SERIES FORECASTING

Classical statistical approaches such as ARIMA (Box et al., 2015) model linear temporal dependencies under stationarity assumptions and remain competitive in low-noise, low-dimensional scenarios. With the advent of deep learning, a broad spectrum of neural architectures has been developed for multivariate time-series forecasting (MTSF), differing primarily in how they capture temporal patterns and inter-variable dependencies. Linear models like DLinear (Zeng et al., 2023) demonstrate that simple channel-wise decomposition of trend and seasonality can already yield strong performance. Attention-based architectures, such as iTransformer (Liu et al., 2024) and TimeXer (Wang et al., 2024a), leverage self-attention to learn adaptive dependencies, with recent designs emphasizing channel-oriented representations to better capture inter-variable correlations.

To more effectively exploit inherent periodicity, period-aware architectures have been introduced. For instance, CycleNet (Lin et al., 2024) injects explicit periodic inductive biases to align model behavior with latent cycles. Nevertheless, two key challenges persist: (*i*) modeling multiple incommensurate periods without phase drift, and (*ii*) jointly learning periodic structure and rich cross-variable dependencies in the presence of noise and nonstationarity. These challenges suggest the need for a more principled framework. In this work, we investigate quantum machine learning as such a framework, leveraging Data-Reuploading Quantum Circuits to encode periodic structures while simultaneously capturing entangled inter-variable relationships.

### 2.2 DATA RE-UPLOADING QUANTUM CIRCUITS AND THE FOURIER PERSPECTIVE

Data re-uploading quantum circuits (DRQC) are quantum machine learning models that repeatedly embed classical data into parameterized quantum circuits, interleaving data-dependent rotations with trainable quantum gates (Schuld et al., 2020; Casas & Cervera-Lierta, 2023). A fundamental theoretical property of DRQC is their equivalence to truncated Fourier series over the encoded variables: under discrete generator spectra and finite depth, the circuit readout corresponds to a finite harmonic expansion whose accessible frequencies are governed by circuit depth and structure. This Fourier characterization clarifies how stacking re-uploading layers systematically expands spectral expressivity and why phase-consistent composition is naturally supported.

Moreover, quantum entanglement operations provide a principled mechanism to couple variables within the circuit state, enabling joint representations that capture cross-variable dependencies

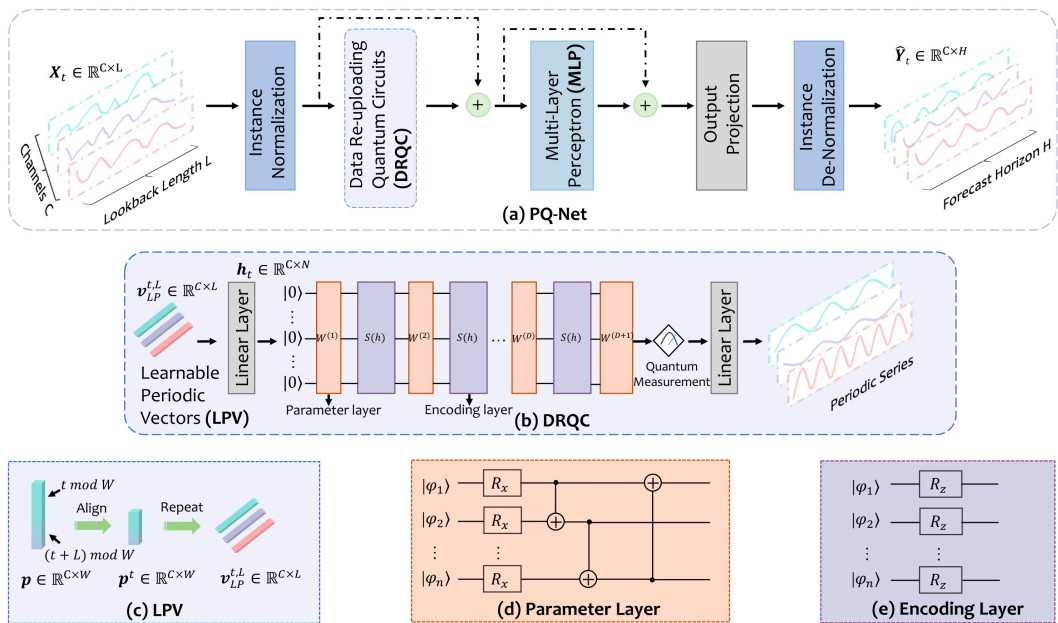

Figure 1: (a) Framework of PQ-Net. It consists mainly of the Learnable Periodic Vectors (LPV) module, the Data-Reuploading Quantum Circuits (DRQC) and Multi-Layer Perceptron (MLP) module, and an Instance Normalization module for mitigating distribution drift. (b) The architecture of the DRQC. (c) The method of generating learned periodic vectors. (d) The quantum circuit of the parameter layer in the DRQC. (e) The quantum circuit of the encoding layer in the DRQC.

alongside periodic structure encoding (Wang et al., 2025; Barthe & Pérez-Salinas, 2024). In the context of MTSF, this dual ability—precise spectral modeling and inter-variable entanglement—directly targets the twin challenges highlighted above. In this work, DRQC serves as the core quantum module of our forecasting architecture, leveraging its Fourier perspective to achieve phase-consistent periodic modeling together with robust multivariate dependency learning.

## 3 METHODOLOGY

Given historical multivariate time series data $\boldsymbol{X}_t \in \mathbb{R}^{C \times L}$, where $C$ denotes the number of variables channels and $L$ the look-back window length, the goal of multivariate time series forecasting (MTSF) is to predict future values $\hat{\boldsymbol{Y}}_t \in \mathbb{R}^{C \times H}$, where $H$ is the forecasting horizon. To this end, we propose the Periodic Quantum Network (PQ-Net), as shown in Figure 1. PQ-Net leverages the expressivity of Data Re-uploading Quantum Circuits (DRQC), whose Fourier-series equivalence enables effective modeling of periodic patterns and cross-channel correlations.

### 3.1 OVERVIEW OF PQ-NET

PQ-Net is a modular forecasting architecture combining normalization, periodic priors, quantum sequence modeling, and feature projection, cf. Figure 1. The pipeline is consisted of Instance Normalization (IN), Learnable Periodic Vectors (LPV), Data-Reuploading Quantum Circuits (DRQC), a lightweight MLP, and Instance De-Normalization.

**Instance Normalization (IN)** To stabilize optimization and enhance cross-channel comparability, we adopt instance normalization (IN) on the input. For input $\boldsymbol{X}_t \in \mathbb{R}^{C \times L}$ with $C$ channels and lookback length $L$, IN standardizes each channel by

$$\boldsymbol{X}'_t = \frac{\boldsymbol{X}_t - \mu}{\sqrt{\sigma^2 + \epsilon}}, \tag{1}$$

where $\mu, \sigma^2 \in \mathbb{R}^C$ are channel-wise mean and variance, and $\epsilon$ ensures numerical stability.

**Learnable Periodic Vectors (LPV)**  To provide phase-aligned periodic priors, we introduce learnable vectors $\boldsymbol{p} \in \mathbb{R}^{C \times W}$ with cycle length $W$. At step $t$, the vector is phase-aligned by shifting $\boldsymbol{p}$ by $t \bmod W$, repeated $\lfloor L/W \rfloor$ times, and truncated to length $L$, the resulting representation is

$$\boldsymbol{v}_{\mathrm{LP}}^{t,L} = \big[ \underbrace{\boldsymbol{p}^t, \ldots, \boldsymbol{p}^t}_{\lfloor L/W \rfloor}, \, \boldsymbol{p}_{[:,0:L \bmod W]}^t \big], \tag{2}$$

which ensures consistent phase anchors across cycles.

**Data-Reuploading Quantum Circuits (DRQC)**  To jointly capture periodic structures and cross-variable dependencies, we employ data-reuploading quantum circuits (DRQC), which enrich the accessible Fourier spectrum through repeated encoding and quantum entanglement. The LPV subsequence is linearly mapped to an embedding vector $\boldsymbol{h}_t \in \mathbb{R}^N$:

$$\boldsymbol{h}_t = W_{\mathrm{lin}} \boldsymbol{v}_{\mathrm{LP}}^{t,L} + \boldsymbol{b}_{\mathrm{lin}}, \tag{3}$$

where $N$ is the number of qubits. The circuit alternates parameter layers:

$$W^i = \bigotimes_{n=1}^{N} R_X(\alpha_{i,n}) \cdot \mathrm{CNOT}_{\mathrm{sym}}, \tag{4}$$

with learnable angles $\alpha_{i,n}$, and encoding layers:

$$S(\boldsymbol{h}_t) = \bigotimes_{n=1}^{N} R_Z(\beta_{i,n} h_{t,n}), \tag{5}$$

with learnable scalings $\beta_{i,n}$. After depth $D$, the final state $\psi^{(D+1)}$ is measured on each qubit:

$$r_n = \langle \psi^{(D+1)} | Z_n | \psi^{(D+1)} \rangle \in [-1, 1], \tag{6}$$

yielding readout $\boldsymbol{r} \in \mathbb{R}^N$, projected by $\boldsymbol{h}_{\mathrm{drqc}} \in \mathbb{R}^{C \times L}$:

$$\boldsymbol{h}_{\mathrm{drqc}} = W_{\mathrm{head}} \boldsymbol{r} + \boldsymbol{b}_{\mathrm{head}}. \tag{7}$$

Repeated data reuploading further enhances the circuit's ability to represent diverse frequency components.

**Prediction Head**  To transform quantum representations into forecasts on the original data scale, we first refine the DRQC features through an MLP and then project them to the forecast horizon $H$, followed by de-normalization to restore the original magnitude. The overall prediction head is formulated as

$$\hat{\boldsymbol{Y}}_t = \Big( W_{\mathrm{out}} \big( W_2 \, \sigma(W_1 \boldsymbol{h}_{\mathrm{drqc}} + \boldsymbol{b}_1) + \boldsymbol{b}_2 \big) + \boldsymbol{b}_{\mathrm{out}} \Big) \odot \sqrt{\sigma^2 + \epsilon} + \mu, \tag{8}$$

where $W_1, W_2, W_{\mathrm{out}}, \boldsymbol{b}_1, \boldsymbol{b}_2, \boldsymbol{b}_{\mathrm{out}}$ are trainable parameters, $\sigma$ is GeLU, and $\odot$ denotes elementwise multiplication.

## 3.2 Fourier-Series Perspective of DRQC

In this section, we establish the connection between DRQC and time–periodicity modeling by showing that the Data-Reuploading Quantum Circuits (DRQC) architecture can be explicitly formulated as a truncated Fourier series.

### 3.2.1 Definition of Quantum Circuit

A parameterized quantum model can be expressed as:

$$\boldsymbol{h} = f_{\boldsymbol{\theta}}(\boldsymbol{h}) = \langle 0 | U^{\dagger}(\boldsymbol{h}, \boldsymbol{\theta}) \, \mathcal{O} \, U(\boldsymbol{h}, \boldsymbol{\theta}) | 0 \rangle, \tag{9}$$

where $|0\rangle$ denotes the initial $N$-qubit state, $U(\boldsymbol{h}, \boldsymbol{\theta})$ is a unitary transformation depending on the classical input $\boldsymbol{h}$ and a (possibly empty) set of trainable parameters $\boldsymbol{\theta}$, and $\mathcal{O}$ is an arbitrary measurement observable. The adjoint $U^{\dagger}$ is the Hermitian conjugate of $U$, and since $U$ is unitary, it

represents its inverse. The model output $f_{\boldsymbol{\theta}}(\mathbf{h})$ is estimated by repeatedly executing the circuit on a quantum device and averaging the measurement results (Schuld et al., 2021).

In DRQC, the circuit $U(\mathbf{h})$ consists of $D$ data reuploading blocks, each composed of a data-encoding layer $S(\cdot)$ and a trainable parameter layer $W(\cdot)$ controlled by $\boldsymbol{\theta}$, cf. Figure 1(d,e). The general structure is:

$$U(\mathbf{h}) = W^{(D+1)} S(\mathbf{h}) W^{(D)} \cdots W^{(2)} S(\mathbf{h}) W^{(1)}, \tag{10}$$

where $S(\mathbf{h})$ applies data-dependent single-qubit gates to encode the classical input vector $\mathbf{h} \in \mathbb{R}^N$ into the quantum state, and $W^{(i)}$ denotes a parameterized multi-qubit unitary (including entangling operations) that is independent of $\mathbf{h}$ and shared across all inputs.

By alternating between $S(\mathbf{h})$ and $W^{(i)}$, the DRQC repeatedly injects the same classical features while transforming the state space through trainable unitaries, thereby enabling rich and structured frequency composition in its output.

### 3.2.2 MATHEMATICAL DERIVATION OF THE FOURIER SERIES FORM

As discussed in the previous section, the repeated data reuploading in DRQC progressively enriches the set of accessible frequencies in the circuit output. Each encoding layer injects LPV-driven phase factors into the quantum state, while each parameter layer mixes these phases across qubits via trainable unitaries and symmetric entanglers. This structure induces a finite and discrete frequency spectrum, whose composition depends on the number of qubits $N$, the eigenvalues of the generator Hamiltonians, and the reuploading depth $D$. In the following, we formalize this intuition by deriving an explicit frequency-domain representation of DRQC and establishing its equivalence to a truncated Fourier series.

Consider a generic $N$-qubit data reuploading quantum circuit, where each encoding layer has the form

$$S(\mathbf{h}) = \mathrm{e}^{-\mathrm{i}q_1 \mathcal{H}} \otimes \cdots \otimes \mathrm{e}^{-\mathrm{i}q_N \mathcal{H}}, \tag{11}$$

with $\otimes$ denoting the tensor product, $\mathbf{h} = (q_1, \ldots, q_N)$ representing the classical data-dependent parameters, and $\mathcal{H}$ being a $d$-dimensional single-qubit Hamiltonian determined by the quantum gates used. The generator $\mathcal{H}$ can always be diagonalized as

$$\mathcal{H} = V^{\dagger} \Sigma V, \tag{12}$$

where $\Sigma$ is a diagonal operator with eigenvalues $\lambda_1, \ldots, \lambda_d$. Without loss of generality, we assume $\mathcal{H}$ is diagonal, since $V$ and $V^{\dagger}$ can be absorbed into adjacent parameter layers (Lloyd, 1996; Abrams & Lloyd, 1999; Aspuru-Guzik et al., 2005; Harrow et al., 2009). This diagonal assumption ensures that $S(\mathbf{h})$ is also diagonal, enabling separation of the data-dependent exponential terms from the fixed unitaries in each amplitude component of the quantum state $U(\mathbf{h}) |0\rangle$.

Under this assumption, the $z$-th component of the quantum state after $D$ encoding layers can be expressed as

$$[U(\mathbf{h}) |0\rangle]_z = \sum_{j_1 \cdots j_D = 1}^{N} \mathrm{e}^{-\mathrm{i}(\lambda_{j_1} + \cdots + \lambda_{j_D})\mathbf{h}} \cdot W_{z j_D}^{(D+1)} \cdots W_{j_2 j_1}^{(2)} W_{j_1 1}^{(1)}, \tag{13}$$

where $W^{(l)}$ denotes the fixed unitary matrices corresponding to parameter layers. For compactness, we introduce the multi-index $\boldsymbol{J} = (j_1, \ldots, j_D) \in [N]^D$, where $[N]^D$ denotes the set of all $D$-tuples with entries in $\{1, \ldots, N\}$. The cumulative eigenvalue associated with $\boldsymbol{J}$ is $\Lambda_{\boldsymbol{J}} = \lambda_{j_1} + \cdots + \lambda_{j_D}$. Then, Eq. (13) can be rewritten as

$$[U(\mathbf{h}) |0\rangle]_z = \sum_{\boldsymbol{J} \in [N]^D} \mathrm{e}^{-\mathrm{i}\Lambda_{\boldsymbol{J}}\mathbf{h}} W_{z j_D}^{(D+1)} \cdots W_{j_2 j_1}^{(2)} W_{j_1 1}^{(1)}. \tag{14}$$

To obtain the output expectation value, we incorporate measurement and the complex conjugate of the above expression, leading to

$$f(\mathbf{h}) = \sum_{\boldsymbol{K}, \boldsymbol{J} \in [N]^D} \mathrm{e}^{\mathrm{i}(\Lambda_{\boldsymbol{K}} - \Lambda_{\boldsymbol{J}})\mathbf{h}} a_{\boldsymbol{K}, \boldsymbol{J}}, \tag{15}$$

where the coefficients $a_{\boldsymbol{K},\boldsymbol{J}}$ depend only on the parameter-layer unitaries and the measurement observable $\mathcal{O}$:

$$a_{\boldsymbol{K},\boldsymbol{J}} = \sum_{z,z'} (W^*)^{(1)}_{1k_1} (W^*)^{(2)}_{j_1 j_2} \cdots (W^*)^{(D+1)}_{j_D z} \mathcal{O}_{z,z'} W^{(D+1)}_{z' j_D} \cdots W^{(2)}_{j_2 j_1} W^{(1)}_{j_1 1}. \tag{16}$$

It is natural to group all terms in Eq. (15) by their shared frequency $\omega = \Lambda_{\boldsymbol{K}} - \Lambda_{\boldsymbol{J}}$, which defines the frequency spectrum accessible to the quantum circuit by

$$\Omega = \{\Lambda_{\boldsymbol{K}} - \Lambda_{\boldsymbol{J}} \mid \boldsymbol{K}, \boldsymbol{J} \in [N]^D\}. \tag{17}$$

Grouping by $\omega \in \Omega$ yields

$$f(\mathbf{h}) = \sum_{\omega \in \Omega} c_\omega \, \mathrm{e}^{\mathrm{i}\omega\mathbf{h}}, \tag{18}$$

where each Fourier coefficient is given by $c_\omega = \sum_{\substack{\boldsymbol{K},\boldsymbol{J}\in[N]^D \\ \Lambda_{\boldsymbol{K}}-\Lambda_{\boldsymbol{J}}=\omega}} a_{\boldsymbol{K},\boldsymbol{J}}$.

Eq. (18) is precisely the form of a (finite) Fourier series, with $\Omega$ as its discrete frequency set and $c_\omega$ as its coefficients.

Therefore, a DRQC with $D$ data reuploading layers and generator eigenvalues $\lambda_i$ implements a truncated Fourier expansion whose spectrum is explicitly determined by Eq. (17). This result establishes a rigorous connection between the circuit architecture and the set of periodic functions it can represent, thereby explaining why DRQC is inherently well-suited for modeling cyclic patterns in multivariate time series.

## 4 EXPERIMENTS

### 4.1 PERIODIC MODELING EXPERIMENTS

Before presenting results on real-world datasets, we first validate the theoretical analysis in Sec. 3.2, which establishes that DRQC can be interpreted as a truncated Fourier series. To this end, we design a controlled synthetic experiment on periodic signals, aiming to directly assess the ability of different architectures to recover oscillatory structures and harmonic components.

#### 4.1.1 SETUP

**Datasets** The synthetic dataset consists of periodic signals with both fundamental oscillatory backbones and higher-order harmonics, designed to test whether models can capture phase consistency and frequency composition. The synthetic data used in this experiment was generated according to Algorithm 1 in Appendix C.1.

**Baselines** We compare DRQC against representative classical function classes, including MLP (Rosenblatt, 1958), KAN (Liu et al., 2025), and Transformer (Vaswani et al., 2017), as well as implicit neural representation models such as ReLU+RFF (Mildenhall et al., 2021) and SIREN (Sitzmann et al., 2020).

#### 4.1.2 RESULTS

As shown in Figure 2, PQ-Net with the DRQC block faithfully reconstructs both the oscillatory backbone and high-frequency harmonics of the target sequence. In contrast, alternative function classes either exhibit phase drift, oversmooth low-frequency trends, or overfit local ripples. These findings provide direct empirical evidence that DRQC excels in capturing periodic structures, thereby supporting its theoretical interpretation as a Fourier-series model. Additional results on synthetic periodic datasets are reported in Appendix E.1.

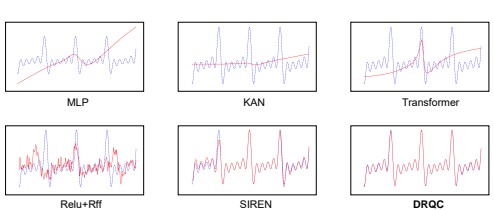

Figure 2: Periodic modeling on synthetic signals. Blue represents actual values, red represents predicted values.

Table 1: Comparisons of multivariate time series forecasting results across 12 real-world datasets. For ETT, Electricity, Solar, Traffic, and Weather datasets, the reported results are averaged over all prediction horizons $H \in \{96, 192, 336, 720\}$. For PEMS datasets, the reported results are averaged over horizons $H \in \{12, 24, 48, 96\}$. Detailed results are available in Table 6. The look-back length $L$ is uniformly fixed at 96. The best results are highlighted in **bold**, the second best are underlined, and the *Count* row counts the number of times each model ranks in the top 2.

| Dataset | PQ-Net (Ours) | | QuLTSF (2025) | | CMoS (2025) | | TimeXer (2024a) | | CycleNet (2024) | | iTransformer (2024) | | MSGNet (2024) | | TimesNet (2023) | | PatchTST (2023) | | Crossformer (2023) | | DLinear (2023) | |
|---|---|---|---|---|---|---|---|---|---|---|---|---|---|---|---|---|---|---|---|---|---|---|
| Metric | MSE↓ | MAE↓ | MSE↓ | MAE↓ | MSE↓ | MAE↓ | MSE↓ | MAE↓ | MSE↓ | MAE↓ | MSE↓ | MAE↓ | MSE↓ | MAE↓ | MSE↓ | MAE↓ | MSE↓ | MAE↓ | MSE↓ | MAE↓ | MSE↓ | MAE↓ |
| ETTh1 | 0.439 | **0.433** | 0.752 | 0.624 | 0.453 | 0.440 | **0.437** | 0.437 | 0.457 | 0.441 | 0.454 | 0.448 | 0.453 | 0.453 | 0.458 | 0.450 | 0.469 | 0.455 | 0.529 | 0.522 | 0.456 | 0.452 |
| ETTh2 | 0.374 | 0.400 | 1.553 | 0.897 | 0.382 | 0.405 | **0.368** | **0.396** | 0.388 | 0.409 | 0.383 | 0.407 | 0.413 | 0.427 | 0.414 | 0.427 | 0.387 | 0.407 | 0.942 | 0.684 | 0.559 | 0.515 |
| ETTm1 | **0.373** | **0.392** | 0.505 | 0.470 | 0.377 | 0.395 | 0.382 | 0.397 | 0.379 | 0.396 | 0.407 | 0.410 | 0.400 | 0.412 | 0.400 | 0.406 | 0.387 | 0.400 | 0.513 | 0.495 | 0.403 | 0.407 |
| ETTm2 | 0.272 | 0.319 | 0.330 | 0.372 | 0.274 | 0.322 | 0.274 | 0.322 | **0.266** | **0.314** | 0.288 | 0.332 | 0.289 | 0.330 | 0.291 | 0.333 | 0.281 | 0.326 | 0.757 | 0.611 | 0.350 | 0.401 |
| Electricity | **0.166** | **0.259** | 0.288 | 0.368 | 0.170 | 0.264 | 0.171 | 0.270 | 0.168 | 0.259 | 0.178 | 0.270 | 0.194 | 0.301 | 0.193 | 0.295 | 0.205 | 0.290 | 0.244 | 0.334 | 0.212 | 0.300 |
| Solar | **0.195** | **0.246** | 0.271 | 0.324 | 0.205 | 0.258 | 0.237 | 0.302 | 0.210 | 0.261 | 0.233 | 0.262 | 0.263 | 0.292 | 0.301 | 0.319 | 0.270 | 0.307 | 0.641 | 0.639 | 0.330 | 0.401 |
| Traffic | 0.453 | 0.275 | 0.781 | 0.441 | 0.469 | 0.295 | 0.466 | 0.287 | 0.472 | 0.301 | **0.428** | 0.282 | 0.660 | 0.382 | 0.620 | 0.336 | 0.481 | 0.300 | 0.550 | 0.304 | 0.625 | 0.383 |
| Weather | 0.241 | 0.269 | 0.256 | 0.301 | 0.245 | 0.273 | 0.241 | 0.271 | 0.243 | 0.271 | 0.258 | 0.278 | 0.249 | 0.278 | 0.259 | 0.287 | 0.259 | 0.273 | 0.259 | 0.315 | 0.265 | 0.317 |
| PEMS03 | **0.090** | **0.192** | 0.189 | 0.296 | 0.119 | 0.226 | 0.112 | 0.214 | 0.118 | 0.226 | 0.113 | 0.222 | 0.150 | 0.251 | 0.147 | 0.248 | 0.180 | 0.291 | 0.169 | 0.282 | 0.278 | 0.375 |
| PEMS04 | **0.090** | **0.194** | 0.194 | 0.303 | 0.117 | 0.230 | 0.105 | 0.209 | 0.111 | 0.232 | 0.111 | 0.221 | 0.122 | 0.239 | 0.129 | 0.241 | 0.195 | 0.307 | 0.209 | 0.314 | 0.295 | 0.388 |
| PEMS07 | **0.078** | **0.173** | 0.324 | 0.378 | 0.110 | 0.211 | 0.085 | 0.182 | 0.113 | 0.214 | 0.101 | 0.204 | 0.122 | 0.227 | 0.125 | 0.226 | 0.211 | 0.303 | 0.235 | 0.315 | 0.329 | 0.396 |
| PEMS08 | **0.140** | **0.220** | 0.298 | 0.312 | 0.148 | 0.243 | 0.175 | 0.250 | 0.150 | 0.246 | 0.150 | 0.226 | 0.205 | 0.285 | 0.193 | 0.271 | 0.280 | 0.321 | 0.268 | 0.307 | 0.379 | 0.416 |
| Average | **0.243** | **0.281** | 0.478 | 0.424 | 0.256 | 0.297 | 0.254 | 0.295 | 0.257 | 0.298 | 0.259 | 0.297 | 0.293 | 0.323 | 0.294 | 0.320 | 0.300 | 0.332 | 0.443 | 0.427 | 0.373 | 0.396 |
| Count | 24 | | 0 | | 6 | | 13 | | 3 | | 2 | | 0 | | 0 | | 0 | | 0 | | 0 | |

## 4.2 MULTIVARIATE TIME SERIES FORECASTING EXPERIMENTS

### 4.2.1 SETUP

In this section, the classical machine learning experiments are implemented using PyTorch (Paszke et al., 2019), while the quantum machine learning modules are developed with PennyLane. Detailed experimental settings are provided in Appendix D.

**Datasets** We evaluate the proposed PQ-Net on 12 widely used real-world datasets, including the ETT series (Zhou et al., 2021), PEMS series (Lin et al., 2024), Electricity, Solar, Traffic, and Weather datasets (Wu et al., 2021). These datasets cover diverse scales, variable dimensionalities, sampling frequencies, and domains. Detailed statistics for each dataset are summarized in Appendix C.2.

**Baselines** To assess the performance of PQ-Net, we compare it against several representative forecasting models from recent years, including CMoS (Si et al., 2025), TimeXer (Wang et al., 2024a), CycleNet (Lin et al., 2024), iTransformer (Liu et al., 2024), MSGNet (Cai et al., 2024), Times-Net (Wu et al., 2023), PatchTST (Nie et al., 2023), Crossformer (Zhang & Yan, 2023), and DLinear (Zeng et al., 2023). It is worth noting that we did not include certain quantum time-series forecasting approaches, such as quantum recurrent neural networks (QRNN) (Li et al., 2023) and quantum long short-term memory networks (QLSTM) (Chen et al., 2022), in our comparisons. Our study primarily focuses on the LSTF model, which employs a multi-step prediction mechanism, making these architectures particularly challenging to reason about even on classical hardware. Furthermore, we found that the results reported in the QuantumTime (Qiao et al., 2025) paper relied on retrospective window settings that we consider unfair, and thus do not provide a suitable comparative baseline (we provide comparative results consistent with the QuantumTime settings in Appendix E.2). Instead, we adopt the recently open-sourced QuLTSF method as our quantum benchmark (Hari et al., 2025). Following the setting of iTransformer, PQ-Net adopts a default look-back window length of 96.

### 4.2.2 MAIN RESULTS

We summarize the forecasting performance of PQ-Net and baselines in Table 1, which reports averaged results across 12 real-world datasets. In terms of ranking statistics, PQ-Net is the only method with top-2 performance across all datasets, with a total of 24 top-2 entries, substantially surpassing the second-best method TimeXer (13 entries). These results highlight that PQ-Net not only pushes forward the frontier of hybrid quantum-classical time series forecasting but also establishes a strong and generalizable benchmark across diverse domains, particularly excelling in high-dimensional multivariate forecasting.

A particularly important observation is that PQ-Net demonstrates clear advantages on high-dimensional multivariate datasets such as Electricity and PEMS, which involve over one hundred correlated variables. These scenarios are notoriously challenging due to the intricate inter-variable dependencies and noise interference. While attention-based models such as iTransformer and TimeXer show competitive performance, their reliance on pure self-attention mechanisms makes

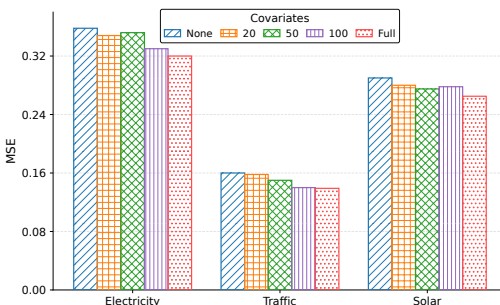

Figure 3: Performance of PQ-Net with varying amounts of covariate information. The forecasting target is the last channel of the dataset, and the results are averaged across forecasting horizons $H \in \{96, 192, 336, 720\}$.

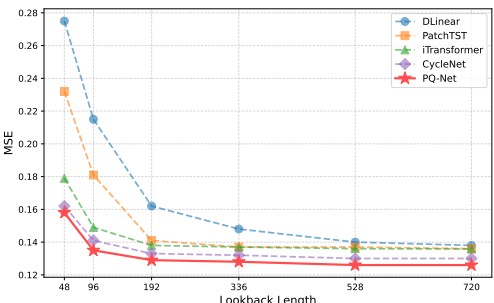

Figure 4: Performance of PQ-Net and comparative models with different look-back lengths on Electricity. The forecast horizon is fixed at 96.

them more vulnerable to noisy disturbances. Conversely, CI-based models like PatchTST and DLinear underperform as they lack explicit mechanisms for capturing cross-variable dependencies. In contrast, PQ-Net leverages its quantum–classical hybrid architecture to model complex correlations more effectively, achieving state-of-the-art performance in both accuracy and robustness on high-dimensional settings.

### 4.2.3 ABLATION STUDIES AND ANALYSIS

**Ablation Studies on PQ-Net Components** To disentangle the contributions of individual components in PQ-Net, we perform ablations on representative high-dimensional datasets. As shown in Table 2, removing LPV leads to the largest degradation, confirming the importance of incorporating explicit periodic priors. Excluding DRQC also causes notable accuracy drops, underscoring its role in stabilizing cross-variable dependencies. The absence of IN results in moderate deterioration,

Table 2: Ablation of PQ-Net on Electricity, PEMS03, and PEMS04. MSE/MAE ($\downarrow$) averaged over $H$, $L$=96. Best in **bold**, second-best underlined.

| IN | LPV | DRQC | MLP | Electricity | PEMS03 | PEMS04 |
|----|-----|------|-----|-------------|--------|--------|
| ✓ | ✓ | ✓ | ✓ | **0.166 / 0.259** | **0.090 / 0.192** | **0.078 / 0.173** |
| ✗ | ✓ | ✓ | ✓ | 0.167 / 0.262 | 0.096 / 0.199 | 0.091 / 0.197 |
| ✓ | ✗ | ✓ | ✓ | 0.174 / 0.267 | 0.112 / 0.220 | 0.113 / 0.223 |
| ✓ | ✓ | ✗ | ✓ | 0.170 / 0.263 | 0.102 / 0.203 | 0.098 / 0.208 |
| ✗ | ✗ | ✗ | ✓ | 0.190 / 0.276 | 0.135 / 0.241 | 0.151 / 0.258 |

showing that normalization improves robustness under non-stationarity. Overall, these results highlight that PQ-Net's strong forecasting performance emerges from the complementary effects of LPV, DRQC, and IN.

**Dependency Studies** Beyond component-level ablations, we further examine whether PQ-Net effectively exploits multivariate dependencies for improved prediction. To this end, we conduct a multivariate-to-univariate forecasting task, where the objective is to predict a single target channel while progressively varying the number of covariates, ranging from none to all available variables. As shown in Figure 3, incorporating even a moderate number of covariates substantially improves predictive accuracy, and performance continues to increase as more covariates are included. These results provide direct evidence that PQ-Net captures robust cross-variable dependencies, which serve as a key factor underlying its forecasting improvements.

**Effect of Look-back Length** The look-back length $L$ governs the richness of historical context available for prediction. In principle, larger $L$ benefits models that can effectively capture long-term dependencies. As illustrated in Figure 4, competitive baselines such as CycleNet (Lin et al., 2024), iTransformer(Liu et al., 2024), PatchTST (Nie et al., 2023), and DLinear (Zeng et al., 2023) all improve with longer look-back windows, indicating their ability to exploit extended temporal information. Notably, PQ-Net exhibits a more pronounced gain as $L$ increases, owing to its DRQC-based representation of periodic structures and its capacity to align with dominant cycles. This demonstrates that PQ-Net not only leverages short-term dynamics but also excels in modeling long-term temporal dependencies when sufficient historical context is provided.

**Hyperparameter Sensitivity** On the Electricity dataset, we ablate three hyperparameters: LPV length $W$, qubit number $n$, and DRQC depth $D$ (Table 3). Aligning $W$ with weekly seasonality

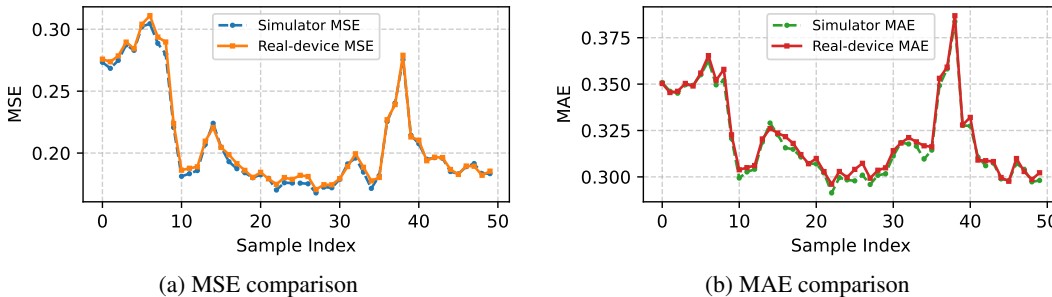

(a) MSE comparison          (b) MAE comparison

Figure 5: Comparison between real-device and simulator results on ETTh1 with 8 qubits. (a) MSE across 50 sequences, (b) MAE across 50 sequences.

Table 3: Ablation study on Electricity dataset with different factors $W$, $n$, and $D$.

| (a) Factor $W$ | | | (b) Factor $n$ | | | (c) Factor $D$ | | |
|---|---|---|---|---|---|---|---|---|
| **Setting** | **MSE↓** | **MAE↓** | **Setting** | **MSE↓** | **MAE↓** | **Setting** | **MSE↓** | **MAE↓** |
| 23 | 0.192 | 0.281 | 4 | 0.184 | 0.274 | 1 | 0.189 | 0.279 |
| **168** | **0.167** | **0.259** | **8** | **0.167** | **0.259** | **3** | **0.167** | **0.259** |
| 336 | 0.171 | 0.263 | 12 | 0.166 | 0.259 | 8 | 0.167 | 0.260 |

(168) yields the lowest error, while non-aligned values degrade performance. Increasing $n$ improves accuracy up to $n = 8$, after which gains saturate. Deeper circuits ($D = 3$) help, but excessive depth offers little benefit. Overall, the setting $W = 168$, $n = 8$, $D = 3$ achieves the best trade-off between accuracy and efficiency.

### 4.3 EXPERIMENTS ON REAL QUANTUM DEVICES

To assess the feasibility of PQ-Net on real quantum hardware, we conducted tests on the `IBM Brisbane` 127-qubit device within the 10-minute free runtime, using the ETTh1 dataset under two representative settings, with all hardware executions run at 128 measurement shots per circuit. First, for a randomly sampled sequence with horizon $96 \rightarrow 192$, we compared different error mitigation levels. Resilience 0 performs no correction, while resilience 1 applies readout error correction and measurement twirling via TREX (Van Den Berg et al., 2022). The obtained MSE/MAE were $0.344/0.386$ and $0.342/0.385$, respectively, nearly identical to the simulator result of $0.342/0.385$. This shows that error mitigation only marginally improves accuracy in this setting. Second, we evaluated 50 sequences with horizon $96 \rightarrow 96$ using both simulator and real device (resilience 0). The simulator averaged $0.208/0.319$, while the device achieved $0.200/0.315$. Detailed per-sequence results in Fig. 5 reveal consistent trends and tight alignment between hardware and simulation, demonstrating that PQ-Net can be reliably executed on near-term quantum devices. Given the good performance on real devices, we plan to conduct more extensive experiments on real quantum devices in the future to further verify the scalability and robustness.

In addition, due to space limitations for the initial submission, the inference time analysis is shown in the Appendix E.3.

## 5 CONCLUSIONS

This paper tackles the challenge of capturing periodic structures and cross-variable dependencies in multivariate time-series forecasting. We propose PQ-Net, a quantum framework built on Data-Reuploading Quantum Circuits (DRQC), which encode periodicity and model variable interactions via quantum entanglement. We theoretically show that DRQC correspond to truncated Fourier expansions, offering a principled and interpretable basis for phase-consistent frequency modeling. Experiments on synthetic and real-world datasets demonstrate that PQ-Net preserves amplitude and phase while handling transients, outperforming period-aware and attention-based baselines. Furthermore, preliminary hardware experiments validate the practicality of PQ-Net, showing competitive performance on real quantum devices.

ETHICS STATEMENT

This work does not involve human subjects, personally identifiable information, or sensitive data. All datasets used are publicly available, properly cited, and comply with their respective licenses. The proposed methodology is designed for time-series forecasting research and does not introduce foreseeable risks of misuse or harmful societal impact. We have adhered to the ICLR Code of Ethics throughout the research and preparation of this paper.

REPRODUCIBILITY STATEMENT

To facilitate reproducibility of PQ-Net, we provide comprehensive dataset descriptions in Appendix C and full experimental details in Appendix D. The supplementary materials include the complete source code, training scripts, and step-by-step instructions, covering data preprocessing, model configurations, and hyperparameters—enabling independent verification of all reported results.

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

## APPENDIX SUMMARY

## A  PRELIMINARIES OF QUANTUM COMPUTING

**Single-Qubit Quantum State.**   The fundamental unit of quantum information is the qubit, which extends the classical bit by allowing superposition. A single qubit state can be expressed as

$$|\psi\rangle = \alpha_0|0\rangle + \alpha_1|1\rangle, \tag{19}$$

where $|0\rangle$ and $|1\rangle$ denote the computational basis states and $\alpha_0, \alpha_1 \in \mathbb{C}$ with $|\alpha_0|^2 + |\alpha_1|^2 = 1$. Measurement collapses the qubit to either $|0\rangle$ or $|1\rangle$ with probability $|\alpha_0|^2$ or $|\alpha_1|^2$. Geometrically, single-qubit states can be visualized on the Bloch sphere, where global phase is ignored and each point corresponds to a valid pure state.

**Multi-Qubit Quantum State.**   A system of $N$ qubits resides in a $2^N$-dimensional Hilbert space. For instance, a two-qubit state is

$$|\phi\rangle = \alpha_{00}|00\rangle + \alpha_{01}|01\rangle + \alpha_{10}|10\rangle + \alpha_{11}|11\rangle, \tag{20}$$

with normalization $\sum_{i,j} |\alpha_{ij}|^2 = 1$. In general, multi-qubit states are constructed via tensor products of individual qubits.

**Quantum Circuits.**   Quantum circuits manipulate qubits through unitary gates, analogous to logic gates in classical computing. Typical single-qubit gates include the Pauli matrices:

$$\sigma_x = \begin{pmatrix} 0 & 1 \\ 1 & 0 \end{pmatrix}, \quad \sigma_y = \begin{pmatrix} 0 & -i \\ i & 0 \end{pmatrix}, \quad \sigma_z = \begin{pmatrix} 1 & 0 \\ 0 & -1 \end{pmatrix}. \tag{21}$$

Two-qubit gates, such as the controlled-NOT (CNOT), enable entanglement by flipping the target qubit conditional on the control qubit being in state $|1\rangle$. A general circuit applies a sequence of such unitary transformations to evolve the input state.

**Parameterized Quantum Circuits (PQC).**   Parameterized quantum circuits form the backbone of many quantum machine learning models. They consist of layers of parameterized single- and two-qubit gates, where tunable parameters (e.g., rotation angles) are optimized during training. The PQC output can be measured to produce classical features, which are then used to define a task-specific loss function.

**Quantum Machine Learning.**   By embedding classical data into PQCs and optimizing gate parameters with respect to a learning objective, quantum circuits can serve as expressive hypothesis classes. This framework underpins the design of hybrid models that integrate quantum feature extraction with classical optimization.

## B CHALLENGES IN QUANTUM NEURAL NETWORKS

Quantum Neural Networks (QNNs) offer a promising paradigm for leveraging quantum computing in machine learning tasks. However, several significant challenges limit their practical adoption:

- **Hardware Constraints**: Despite rapid advancements, today's quantum hardware is still in the Noisy Intermediate-Scale Quantum (NISQ) era. The number of available qubits is limited, and they are highly susceptible to noise and decoherence. This makes it difficult to deploy deep or large-scale quantum models, as the circuits either become too noisy to produce meaningful results or exceed hardware limits. Moreover, the short coherence time imposes tight constraints on circuit depth and execution time.

- **Computational Overhead**: Due to the lack of large-scale fault-tolerant quantum computers, most quantum machine learning models, including PQ-Net, must be simulated on classical hardware. Simulating quantum systems grows exponentially with the number of qubits, resulting in high memory and time complexity. As a consequence, both the training and inference phases of quantum neural networks are computationally expensive, making large-scale experimentation infeasible.

- **Optimization Difficulties**: Quantum neural networks often face the problem of barren plateaus, where the gradient of the loss function vanishes exponentially with the number of qubits or the depth of the circuit. This severely hampers the ability to train the network using gradient-based methods. Furthermore, the loss landscapes of quantum models are often non-convex and exhibit complex behavior, leading to instability during training and a high sensitivity to initialization and learning rates.

## C DETAILS OF DATASETS

Table 4: Detailed information about the datasets. The hyperparameter $W$ of LPV is configured to align with the stable cycle length of each dataset, following the guidelines in CycleNet Lin et al. (2024).

| Dataset | Channels | Timesteps | Interval | HParam. $W$ | Domain |
|---|---|---|---|---|---|
| ETTh1 | 7 | 14,400 | 1 hour | 24 | Electricity |
| ETTh2 | 7 | 14,400 | 1 hour | 24 | Electricity |
| ETTm1 | 7 | 57,600 | 15 mins | 96 | Electricity |
| ETTm2 | 7 | 57,600 | 15 mins | 96 | Electricity |
| Electricity | 321 | 26,304 | 1 hour | 168 | Electricity |
| Solar | 137 | 52,560 | 10 mins | 144 | Energy |
| Traffic | 862 | 17,544 | 1 hour | 168 | Transportation |
| Weather | 21 | 52,696 | 10 mins | 144 | Weather |
| PEMS03 | 358 | 26,208 | 5 mins | 288 | Transportation |
| PEMS04 | 307 | 16,992 | 5 mins | 288 | Transportation |
| PEMS07 | 883 | 28,224 | 5 mins | 288 | Transportation |
| PEMS08 | 170 | 17,856 | 5 mins | 288 | Transportation |

### C.1 DETAILS OF PERIODIC MODELING DATASETS

The synthetic data generation algorithm for the periodic modeling experiment is shown in Algorithm 1.

### C.2 DETAILS OF TIME SERIES FORECASTING DATASETS

Detailed statistics for each dataset are summarized in Table 4. The benchmark suite covers four major domains—Electricity, Energy, Transportation, and Weather—providing a diverse set of temporal dynamics and scales for evaluation.

---

**Algorithm 1** generate periodic data.

---

```
degree = 2 # degree of the target function
scaling = 1 # scaling of the data
coeffs = [0.15 + 0.15j] * degree # coefficients of non-zero frequencies
coeff0 = 0.1 # coefficient of zero frequency

def target_function(x):
    """Generate a truncated Fourier series, where the data gets re-scaled."""

    res = coeff0
    for idx, coeff in enumerate(coeffs):
        exponent = np.complex128(scaling * (idx + 1) * x * 1j)
        conj_coeff = np.conjugate(coeff)
        res += coeff * np.exp(exponent) + conj_coeff * np.exp(-exponent)
    return np.real(res)

t = np.linspace(-6, 6, 600)
data = np.array([target_function(x_) for x_ in t])
```

---

**ETT Series.** The ETTh1, ETTh2, ETTm1, and ETTm2 datasets are collected from electricity transformers in China (Zhou et al., 2021). They contain load and voltage measurements at either hourly or 15-minute resolution, and are widely used to benchmark long sequence time-series forecasting. The cycle length $W$ is aligned with daily or weekly periodicities (24 or 96 timesteps).

**Electricity.** The Electricity dataset records hourly electricity consumption of 321 clients. The data span several years and exhibit strong daily and weekly seasonalities. We set $W = 168$ to capture the weekly cycle.

**Solar and Weather.** The Solar dataset consists of 137 solar power plant outputs sampled every 10 minutes, while the Weather dataset contains 21 meteorological variables (e.g., temperature, humidity, wind speed) recorded every 10 minutes. Both datasets show complex periodic structures, and we configure $W = 144$ to match the daily cycle.

**Traffic.** The Traffic dataset measures road occupancy rates from 862 sensors on the San Francisco Bay Area freeways, sampled every hour. This dataset is characterized by strong weekly patterns, for which we set $W = 168$.

**PEMS Series.** The PEMS03, PEMS04, PEMS07, and PEMS08 datasets are large-scale traffic datasets collected from the Caltrans Performance Measurement System (PEMS). They record 5-minute aggregated traffic speed readings from hundreds of sensors. Following CycleNet (Lin et al., 2024), we set $W = 288$, corresponding to a daily cycle (24 hours × 12 samples per hour).

Overall, these datasets span a wide range of temporal resolutions (from 5 minutes to 1 hour), sequence lengths, and domain-specific periodicities. Such diversity ensures a comprehensive evaluation of forecasting models under different noise levels, seasonal patterns, and cross-channel dependencies.

## D EXPERIMENTAL DETAILS

The complete experimental setup is available in our supplementary materials. All experiments are implemented in PyTorch (Paszke et al., 2019) and executed on a single NVIDIA GeForce RTX 4090 GPU with 24 GB memory. Training is performed using the Adam optimizer (Kingma, 2014) with the L2 loss function. Dataset splits follow common practice in prior works such as iTransformer (Liu et al., 2024) and TimesNet (Wu et al., 2023): we adopt a 6:2:2 ratio for the ETT and PEMS series datasets, and a 7:1:2 ratio for the remaining datasets.

PQ-Net is trained for 30 epochs with early stopping, where training halts if the validation loss does not improve for 5 consecutive epochs. The learning rate is set to $3 \times 10^{-3}$ for most datasets and reduced to $1 \times 10^{-3}$ for smaller datasets (e.g., the ETT series). Batch sizes are adjusted according to dataset scale to balance GPU utilization and memory efficiency (e.g., 16 for Traffic, 64 for Weather).

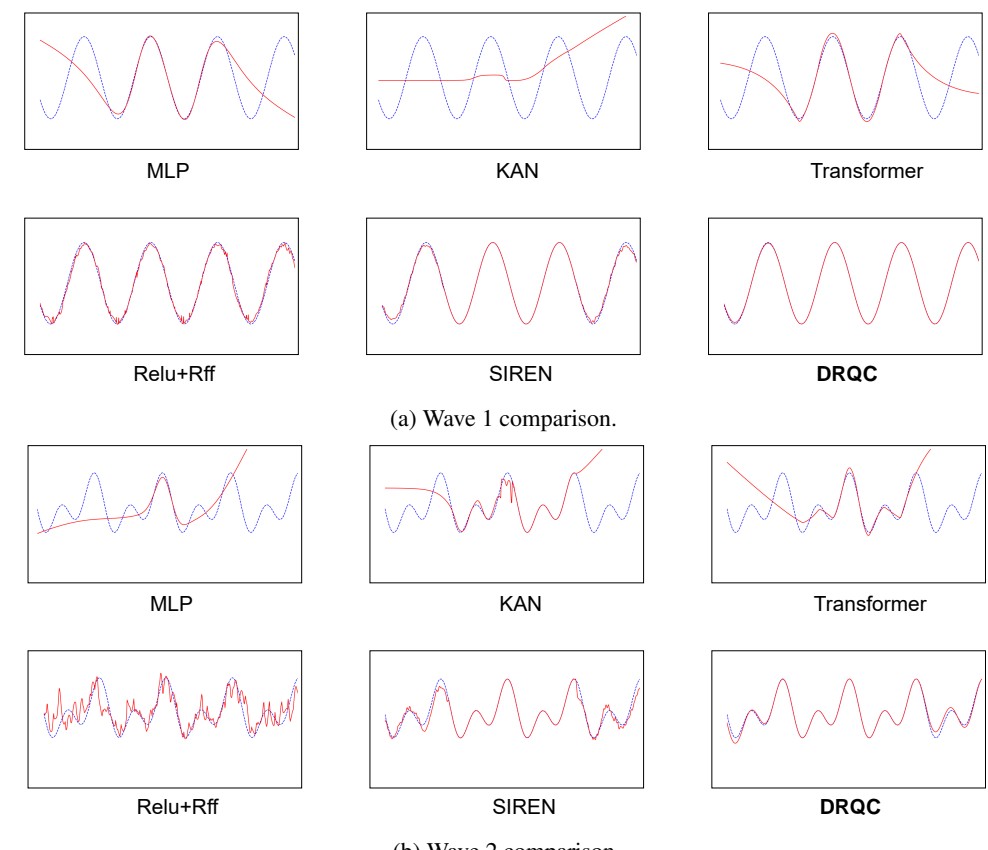

(a) Wave 1 comparison.

(b) Wave 2 comparison.

Figure 6: Comparisons in periodicity modeling for two cases.

For model hyperparameters, the number of qubits is fixed at 8 unless otherwise specified, the depth of DRQC layers is set to 3, and the LPV prior length $W$ is aligned with the dominant cycle of each dataset, as summarized in Table 4. Dropout is applied with a default rate of 0.5 in hidden layers; in the output layer, the dropout rate is set to 0.5 for smaller datasets (e.g., ETT and Weather) and 0 for larger ones. To ensure reproducibility, all experiments are repeated with three seeds (2023, 2024, 2025), and the average performance across runs is reported.

The quantum modules of PQ-Net are implemented with the PennyLane framework (Bergholm et al., 2018), which provides a hybrid quantum-classical interface. All DRQC circuits are executed on the `default.qubit` backend, and gradients are computed using the parameter-shift rule, allowing end-to-end training with PyTorch optimizers. This integration ensures that PQ-Net can be trained efficiently in a fully differentiable manner, while faithfully capturing the Fourier-based expressivity of the quantum circuits.

# E ADDITIONAL EXPERIMENTS

## E.1 SYNTHETIC EXPERIMENTS

Figure 6 is the periodicity modeling results for the other two frequency waveforms mentioned in the main text, and it can be seen that DRQC maintains excellent performance at different frequencies.

## E.2 TIME SERIES FORECASTING EXPERIMENTS

**Full Results of Time Series Forecasting Experiments**  Table 6 reports the full multivariate forecasting results averaged over standard horizons for each dataset. Overall, **PQ-Net** consistently delivers state-of-the-art performance: it attains the **best** average MSE/MAE on 8/12 datasets

Table 5: Comparisons between PQ-Net and QuantumTime on Traffic and Weather datasets across different forecasting horizons $H \in \{96, 192, 336, 720\}$. The best results are highlighted in **bold**.

| Dataset | $H$ | PQ-Net (Ours) | | QuantumTime | |
|---|---|---|---|---|---|
| | | MSE↓ | MAE↓ | MSE↓ | MAE↓ |
| Traffic | 96 | **0.379** | **0.265** | 0.387 | 0.273 |
| | 192 | **0.393** | **0.272** | 0.399 | 0.276 |
| | 336 | **0.405** | **0.277** | 0.413 | 0.286 |
| | 720 | **0.438** | **0.294** | 0.446 | 0.301 |
| | Avg | **0.404** | **0.277** | 0.411 | 0.284 |
| Weather | 96 | **0.150** | **0.200** | 0.166 | 0.223 |
| | 192 | **0.189** | **0.242** | 0.209 | 0.263 |
| | 336 | **0.240** | **0.281** | 0.252 | 0.300 |
| | 720 | **0.298** | **0.331** | 0.299 | 0.338 |
| | Avg | **0.219** | **0.264** | 0.232 | 0.281 |

Table 6: Full multivariate time series forecasting results of Table 1. For ETT, Electricity, Solar, Traffic, and Weather datasets, the reported results are averaged over all prediction horizons $H \in \{96, 192, 336, 720\}$. For PEMS datasets, the reported results are averaged over horizons $H \in \{12, 24, 48, 96\}$. The look-back length $L$ is fixed at 96. The reproduced baseline results are sourced from QuLTSF (Hari et al., 2025), CMoS (Si et al., 2025), TimeXer (Wang et al., 2024a), iTransformer (Liu et al., 2024), and CycleNet (Lin et al., 2024). The best results are highlighted in **bold**, while the second-best results are underlined.

| Model | | PQ-Net (Ours) | | QuLTSF (2025) | | CMoS (2025) | | TimeXer (2024a) | | CycleNet (2024) | | iTransformer (2024) | | MSGNet (2024) | | TimesNet (2023) | | PatchTST (2023) | | Crossformer (2023) | | DLinear (2023) | |
|---|---|---|---|---|---|---|---|---|---|---|---|---|---|---|---|---|---|---|---|---|---|---|---|
| Metric | | MSE↓ | MAE↓ | MSE↓ | MAE↓ | MSE↓ | MAE↓ | MSE↓ | MAE↓ | MSE↓ | MAE↓ | MSE↓ | MAE↓ | MSE↓ | MAE↓ | MSE↓ | MAE↓ | MSE↓ | MAE↓ | MSE↓ | MAE↓ | MSE↓ | MAE↓ |
| ETTh1 | 96 | **0.368** | **0.392** | 0.711 | 0.592 | 0.376 | 0.397 | 0.382 | 0.403 | 0.375 | 0.395 | 0.386 | 0.405 | 0.390 | 0.411 | 0.384 | 0.402 | 0.414 | 0.419 | 0.423 | 0.448 | 0.386 | 0.400 |
| | 192 | **0.423** | **0.422** | 0.737 | 0.606 | 0.434 | 0.430 | 0.429 | 0.435 | 0.436 | 0.428 | 0.441 | 0.436 | 0.443 | 0.442 | 0.436 | 0.429 | 0.460 | 0.445 | 0.471 | 0.474 | 0.437 | 0.432 |
| | 336 | 0.474 | **0.443** | 0.746 | 0.622 | 0.488 | 0.450 | 0.468 | 0.448 | 0.496 | 0.455 | 0.487 | 0.458 | 0.482 | 0.469 | 0.491 | 0.469 | 0.501 | 0.466 | 0.570 | 0.546 | 0.481 | 0.459 |
| | 720 | 0.490 | 0.473 | 0.813 | 0.677 | 0.515 | 0.482 | 0.469 | 0.461 | 0.520 | 0.484 | 0.503 | 0.491 | 0.496 | 0.488 | 0.521 | 0.500 | 0.500 | 0.488 | 0.653 | 0.621 | 0.519 | 0.516 |
| | Avg | 0.439 | 0.433 | 0.752 | 0.624 | 0.453 | 0.440 | 0.437 | 0.437 | 0.457 | 0.441 | 0.454 | 0.448 | 0.453 | 0.453 | 0.458 | 0.450 | 0.469 | 0.455 | 0.529 | 0.522 | 0.456 | 0.452 |
| ETTh2 | 96 | 0.293 | 0.341 | 1.394 | 0.841 | 0.295 | 0.345 | **0.286** | **0.338** | 0.298 | 0.344 | 0.297 | 0.349 | 0.329 | 0.371 | 0.340 | 0.374 | 0.302 | 0.348 | 0.745 | 0.584 | 0.333 | 0.387 |
| | 192 | 0.366 | 0.388 | 1.505 | 0.881 | 0.370 | 0.392 | 0.363 | 0.389 | 0.372 | 0.396 | 0.380 | 0.400 | 0.402 | 0.414 | 0.402 | 0.414 | 0.388 | 0.400 | 0.877 | 0.656 | 0.477 | 0.476 |
| | 336 | 0.416 | 0.427 | 1.537 | 0.895 | 0.429 | 0.436 | 0.414 | 0.423 | 0.431 | 0.439 | 0.428 | 0.432 | 0.440 | 0.445 | 0.452 | 0.452 | 0.426 | 0.433 | 1.043 | 0.731 | 0.594 | 0.541 |
| | 720 | 0.423 | 0.442 | 1.774 | 0.972 | 0.432 | 0.448 | 0.408 | 0.432 | 0.450 | 0.458 | 0.427 | 0.445 | 0.480 | 0.477 | 0.462 | 0.468 | 0.431 | 0.446 | 1.104 | 0.763 | 0.831 | 0.657 |
| | Avg | 0.374 | 0.400 | 1.553 | 0.897 | 0.382 | 0.405 | 0.368 | 0.396 | 0.388 | 0.409 | 0.383 | 0.407 | 0.413 | 0.427 | 0.414 | 0.427 | 0.387 | 0.407 | 0.942 | 0.684 | 0.559 | 0.515 |
| ETTm1 | 96 | **0.309** | **0.352** | 0.466 | 0.444 | 0.317 | 0.358 | 0.318 | 0.356 | 0.319 | 0.360 | 0.334 | 0.368 | 0.319 | 0.366 | 0.338 | 0.375 | 0.329 | 0.367 | 0.404 | 0.426 | 0.345 | 0.372 |
| | 192 | **0.355** | **0.378** | 0.475 | 0.451 | 0.359 | 0.380 | 0.362 | 0.383 | 0.360 | 0.381 | 0.377 | 0.391 | 0.377 | 0.397 | 0.374 | 0.387 | 0.367 | 0.385 | 0.450 | 0.451 | 0.380 | 0.389 |
| | 336 | **0.386** | **0.401** | 0.521 | 0.479 | 0.388 | 0.403 | 0.395 | 0.407 | 0.389 | 0.403 | 0.426 | 0.420 | 0.417 | 0.422 | 0.410 | 0.411 | 0.399 | 0.410 | 0.532 | 0.515 | 0.413 | 0.413 |
| | 720 | **0.441** | **0.436** | 0.559 | 0.506 | 0.445 | 0.439 | 0.452 | 0.441 | 0.447 | 0.441 | 0.491 | 0.459 | 0.487 | 0.463 | 0.478 | 0.450 | 0.454 | 0.439 | 0.666 | 0.589 | 0.474 | 0.453 |
| | Avg | **0.373** | **0.392** | 0.505 | 0.470 | 0.377 | 0.395 | 0.382 | 0.397 | 0.379 | 0.396 | 0.407 | 0.410 | 0.400 | 0.412 | 0.400 | 0.406 | 0.387 | 0.400 | 0.513 | 0.495 | 0.403 | 0.407 |
| ETTm2 | 96 | 0.170 | 0.254 | 0.191 | 0.284 | 0.171 | 0.257 | 0.171 | 0.256 | **0.163** | **0.246** | 0.180 | 0.264 | 0.182 | 0.266 | 0.187 | 0.267 | 0.175 | 0.259 | 0.287 | 0.366 | 0.193 | 0.292 |
| | 192 | 0.234 | 0.296 | 0.264 | 0.333 | 0.236 | 0.299 | 0.237 | 0.299 | **0.229** | **0.290** | 0.250 | 0.309 | 0.248 | 0.306 | 0.249 | 0.309 | 0.241 | 0.302 | 0.414 | 0.492 | 0.284 | 0.362 |
| | 336 | 0.292 | 0.333 | 0.378 | 0.402 | 0.294 | 0.337 | 0.296 | 0.338 | **0.284** | **0.327** | 0.311 | 0.348 | 0.312 | 0.346 | 0.321 | 0.351 | 0.305 | 0.343 | 0.597 | 0.542 | 0.369 | 0.427 |
| | 720 | 0.392 | 0.393 | 0.486 | 0.468 | 0.393 | 0.395 | 0.392 | 0.394 | **0.389** | **0.391** | 0.412 | 0.407 | 0.414 | 0.404 | 0.408 | 0.403 | 0.402 | 0.400 | 1.730 | 1.042 | 0.554 | 0.522 |
| | Avg | 0.272 | 0.319 | 0.330 | 0.372 | 0.274 | 0.322 | 0.274 | 0.322 | **0.266** | **0.314** | 0.288 | 0.332 | 0.289 | 0.330 | 0.291 | 0.333 | 0.281 | 0.326 | 0.757 | 0.611 | 0.350 | 0.401 |
| Electricity | 96 | **0.135** | **0.230** | 0.234 | 0.326 | 0.138 | 0.236 | 0.140 | 0.242 | 0.136 | 0.229 | 0.148 | 0.240 | 0.165 | 0.274 | 0.168 | 0.272 | 0.181 | 0.270 | 0.219 | 0.314 | 0.197 | 0.282 |
| | 192 | 0.153 | **0.246** | 0.264 | 0.348 | 0.156 | 0.252 | 0.157 | 0.256 | 0.152 | 0.244 | 0.162 | 0.253 | 0.185 | 0.292 | 0.184 | 0.289 | 0.188 | 0.274 | 0.231 | 0.322 | 0.196 | 0.285 |
| | 336 | **0.170** | **0.264** | 0.298 | 0.376 | 0.175 | 0.266 | 0.176 | 0.275 | 0.170 | 0.264 | 0.178 | 0.269 | 0.197 | 0.304 | 0.198 | 0.300 | 0.204 | 0.293 | 0.246 | 0.337 | 0.209 | 0.301 |
| | 720 | **0.205** | **0.297** | 0.355 | 0.421 | 0.212 | 0.301 | 0.211 | 0.306 | 0.212 | 0.299 | 0.225 | 0.317 | 0.231 | 0.332 | 0.220 | 0.320 | 0.246 | 0.324 | 0.280 | 0.363 | 0.245 | 0.333 |
| | Avg | **0.166** | **0.259** | 0.288 | 0.368 | 0.170 | 0.264 | 0.171 | 0.270 | 0.168 | 0.259 | 0.178 | 0.270 | 0.194 | 0.301 | 0.193 | 0.295 | 0.205 | 0.290 | 0.244 | 0.334 | 0.212 | 0.300 |
| Solar-Energy | 96 | **0.177** | **0.236** | 0.243 | 0.301 | 0.188 | 0.243 | 0.215 | 0.295 | 0.190 | 0.247 | 0.203 | 0.237 | 0.210 | 0.246 | 0.250 | 0.292 | 0.234 | 0.286 | 0.310 | 0.331 | 0.290 | 0.378 |
| | 192 | **0.194** | **0.250** | 0.270 | 0.330 | 0.198 | 0.264 | 0.236 | 0.301 | 0.210 | 0.266 | 0.233 | 0.261 | 0.265 | 0.290 | 0.296 | 0.318 | 0.267 | 0.310 | 0.734 | 0.725 | 0.320 | 0.398 |
| | 336 | **0.201** | **0.247** | 0.286 | 0.336 | 0.214 | 0.262 | 0.252 | 0.307 | 0.217 | 0.266 | 0.248 | 0.273 | 0.294 | 0.318 | 0.319 | 0.330 | 0.290 | 0.315 | 0.750 | 0.735 | 0.353 | 0.415 |
| | 720 | **0.206** | **0.251** | 0.283 | 0.329 | 0.218 | 0.264 | 0.244 | 0.305 | 0.223 | 0.266 | 0.249 | 0.275 | 0.285 | 0.317 | 0.338 | 0.337 | 0.289 | 0.317 | 0.769 | 0.765 | 0.356 | 0.413 |
| | Avg | **0.195** | **0.246** | 0.271 | 0.324 | 0.205 | 0.258 | 0.237 | 0.302 | 0.210 | 0.261 | 0.233 | 0.262 | 0.263 | 0.292 | 0.301 | 0.319 | 0.270 | 0.307 | 0.641 | 0.639 | 0.330 | 0.401 |
| Traffic | 96 | 0.425 | **0.260** | 0.733 | 0.423 | 0.449 | 0.290 | 0.428 | 0.271 | 0.458 | 0.296 | 0.395 | 0.268 | 0.608 | 0.349 | 0.593 | 0.321 | 0.462 | 0.290 | 0.522 | 0.290 | 0.650 | 0.396 |
| | 192 | 0.445 | **0.270** | 0.762 | 0.430 | 0.452 | 0.291 | 0.448 | 0.282 | 0.457 | 0.294 | 0.417 | 0.276 | 0.634 | 0.371 | 0.617 | 0.336 | 0.466 | 0.290 | 0.530 | 0.293 | 0.598 | 0.370 |
| | 336 | 0.455 | **0.277** | 0.791 | 0.447 | 0.465 | 0.298 | 0.473 | 0.289 | 0.470 | 0.299 | 0.433 | 0.283 | 0.669 | 0.388 | 0.629 | 0.336 | 0.482 | 0.300 | 0.558 | 0.305 | 0.605 | 0.373 |
| | 720 | 0.488 | **0.294** | 0.837 | 0.465 | 0.511 | 0.302 | 0.516 | 0.307 | 0.502 | 0.314 | 0.467 | 0.302 | 0.729 | 0.420 | 0.640 | 0.350 | 0.514 | 0.320 | 0.589 | 0.328 | 0.645 | 0.394 |
| | Avg | 0.453 | **0.275** | 0.781 | 0.441 | 0.469 | 0.295 | 0.466 | 0.287 | 0.472 | 0.301 | 0.428 | 0.282 | 0.660 | 0.382 | 0.620 | 0.336 | 0.481 | 0.300 | 0.550 | 0.304 | 0.625 | 0.383 |
| Weather | 96 | **0.154** | **0.200** | 0.187 | 0.243 | 0.159 | 0.204 | 0.157 | 0.205 | 0.158 | 0.203 | 0.174 | 0.214 | 0.163 | 0.212 | 0.172 | 0.220 | 0.177 | 0.210 | 0.158 | 0.230 | 0.196 | 0.255 |
| | 192 | 0.205 | **0.246** | 0.219 | 0.271 | 0.210 | 0.250 | 0.204 | 0.247 | 0.207 | 0.247 | 0.221 | 0.254 | 0.211 | 0.254 | 0.219 | 0.261 | 0.225 | 0.250 | 0.206 | 0.277 | 0.237 | 0.296 |
| | 336 | **0.262** | **0.288** | 0.271 | 0.318 | 0.264 | 0.291 | 0.261 | 0.290 | 0.262 | 0.289 | 0.278 | 0.296 | 0.273 | 0.299 | 0.280 | 0.306 | 0.278 | 0.290 | 0.272 | 0.335 | 0.283 | 0.335 |
| | 720 | 0.343 | 0.343 | 0.345 | 0.372 | 0.345 | 0.346 | 0.340 | 0.341 | 0.344 | 0.344 | 0.358 | 0.349 | 0.351 | 0.348 | 0.365 | 0.359 | 0.354 | 0.340 | 0.398 | 0.418 | 0.345 | 0.381 |
| | Avg | 0.241 | **0.269** | 0.256 | 0.301 | 0.245 | 0.273 | 0.241 | 0.271 | 0.243 | 0.271 | 0.258 | 0.278 | 0.249 | 0.278 | 0.259 | 0.287 | 0.259 | 0.273 | 0.259 | 0.315 | 0.265 | 0.317 |
| PEMS03 | 12 | **0.058** | **0.157** | 0.093 | 0.203 | 0.067 | 0.173 | 0.070 | 0.173 | 0.066 | 0.172 | 0.071 | 0.174 | 0.078 | 0.187 | 0.085 | 0.192 | 0.090 | 0.216 | 0.090 | 0.203 | 0.122 | 0.243 |
| | 24 | **0.071** | **0.174** | 0.141 | 0.254 | 0.091 | 0.202 | 0.092 | 0.194 | 0.089 | 0.201 | 0.093 | 0.201 | 0.108 | 0.218 | 0.118 | 0.223 | 0.142 | 0.259 | 0.121 | 0.240 | 0.201 | 0.317 |
| | 48 | **0.097** | **0.201** | 0.225 | 0.332 | 0.136 | 0.247 | 0.129 | 0.229 | 0.125 | 0.236 | | | 0.178 | 0.272 | 0.155 | 0.260 | 0.221 | 0.319 | 0.202 | 0.317 | 0.333 | 0.425 |
| | 96 | **0.132** | **0.234** | 0.296 | 0.395 | 0.181 | 0.280 | 0.157 | 0.261 | 0.182 | 0.282 | 0.164 | 0.275 | 0.238 | 0.328 | 0.228 | 0.317 | 0.269 | 0.370 | 0.262 | 0.367 | 0.457 | 0.515 |
| | Avg | **0.090** | **0.192** | 0.189 | 0.296 | 0.119 | 0.226 | 0.112 | 0.214 | 0.118 | 0.226 | 0.118 | 0.222 | 0.150 | 0.251 | 0.147 | 0.248 | 0.180 | 0.291 | 0.169 | 0.282 | 0.278 | 0.375 |
| PEMS04 | 12 | **0.066** | **0.165** | 0.099 | 0.212 | 0.077 | 0.185 | 0.074 | 0.178 | 0.078 | 0.186 | 0.078 | 0.183 | 0.086 | 0.199 | 0.087 | 0.195 | 0.105 | 0.224 | 0.098 | 0.218 | 0.148 | 0.272 |
| | 24 | **0.076** | **0.180** | 0.149 | 0.266 | 0.097 | 0.210 | 0.087 | 0.195 | 0.099 | 0.212 | 0.095 | 0.205 | 0.101 | 0.218 | 0.103 | 0.215 | 0.153 | 0.275 | 0.131 | 0.256 | 0.224 | 0.340 |
| | 48 | **0.095** | **0.202** | 0.233 | 0.341 | 0.130 | 0.246 | 0.110 | 0.214 | 0.133 | 0.248 | 0.120 | 0.233 | 0.127 | 0.247 | 0.136 | 0.250 | 0.229 | 0.339 | 0.205 | 0.326 | 0.355 | 0.437 |
| | 96 | **0.121** | **0.229** | 0.296 | 0.392 | 0.165 | 0.279 | 0.148 | 0.251 | 0.167 | 0.281 | 0.150 | 0.262 | 0.174 | 0.292 | 0.190 | 0.303 | 0.291 | 0.389 | 0.402 | 0.457 | 0.452 | 0.504 |
| | Avg | **0.090** | **0.194** | 0.194 | 0.303 | 0.117 | 0.230 | 0.105 | 0.209 | 0.119 | 0.232 | 0.111 | 0.221 | 0.122 | 0.239 | 0.129 | 0.241 | 0.195 | 0.307 | 0.209 | 0.314 | 0.295 | 0.388 |
| PEMS07 | 12 | **0.053** | **0.144** | 0.111 | 0.207 | 0.060 | 0.165 | 0.057 | 0.152 | 0.062 | 0.162 | 0.067 | 0.165 | 0.079 | 0.183 | 0.082 | 0.181 | 0.095 | 0.207 | 0.094 | 0.200 | 0.115 | 0.242 |
| | 24 | **0.066** | **0.160** | 0.205 | 0.312 | 0.084 | 0.189 | 0.079 | 0.179 | 0.086 | 0.192 | 0.088 | 0.190 | 0.099 | 0.206 | 0.101 | 0.204 | 0.150 | 0.262 | 0.139 | 0.247 | 0.210 | 0.329 |
| | 48 | **0.086** | **0.183** | 0.389 | 0.446 | 0.124 | 0.230 | 0.099 | 0.191 | 0.128 | 0.234 | 0.110 | 0.215 | 0.133 | 0.239 | 0.134 | 0.238 | 0.253 | 0.340 | 0.311 | 0.369 | 0.398 | 0.458 |
| | 96 | **0.105** | **0.205** | 0.590 | 0.545 | 0.172 | 0.265 | 0.107 | 0.205 | 0.176 | 0.268 | 0.139 | 0.245 | 0.179 | 0.279 | 0.181 | 0.279 | 0.346 | 0.404 | 0.396 | 0.442 | 0.594 | 0.553 |
| | Avg | **0.078** | **0.173** | 0.324 | 0.378 | 0.110 | 0.211 | 0.085 | 0.182 | 0.113 | 0.214 | 0.101 | 0.204 | 0.122 | 0.227 | 0.125 | 0.226 | 0.211 | 0.303 | 0.235 | 0.315 | 0.329 | 0.396 |
| PEMS08 | 12 | **0.072** | **0.171** | 0.188 | 0.215 | 0.080 | 0.182 | 0.075 | 0.176 | 0.082 | 0.185 | 0.079 | 0.182 | 0.105 | 0.211 | 0.112 | 0.212 | 0.168 | 0.232 | 0.165 | 0.214 | 0.154 | 0.276 |
| | 24 | **0.098** | **0.197** | 0.243 | 0.268 | 0.114 | 0.222 | 0.102 | 0.201 | 0.117 | 0.226 | 0.115 | 0.219 | 0.141 | 0.243 | 0.141 | 0.238 | 0.224 | 0.281 | 0.215 | 0.260 | 0.248 | 0.353 |
| | 48 | **0.153** | 0.242 | 0.334 | 0.348 | 0.165 | 0.263 | 0.158 | 0.248 | 0.169 | 0.268 | 0.186 | 0.235 | 0.211 | 0.300 | 0.198 | 0.283 | 0.321 | 0.354 | 0.315 | 0.355 | 0.440 | 0.470 |
| | 96 | 0.235 | 0.271 | 0.426 | 0.418 | 0.231 | 0.304 | 0.366 | 0.377 | 0.233 | 0.306 | **0.221** | **0.267** | 0.364 | 0.387 | 0.320 | 0.351 | 0.408 | 0.417 | 0.377 | 0.397 | 0.674 | 0.565 |
| | Avg | **0.140** | **0.220** | 0.298 | 0.312 | 0.148 | 0.243 | 0.175 | 0.250 | 0.150 | 0.246 | 0.150 | 0.226 | 0.205 | 0.285 | 0.193 | 0.271 | 0.280 | 0.321 | 0.268 | 0.307 | 0.379 | 0.416 |

(ETTm1, Electricity, Solar-Energy, Weather, PEMS03/04/07/08), achieves the **best MAE** and

second-best MSE on ETTh1 and Traffic, and ranks second on both metrics for ETTh2 and ETTm2. In total, PQ-Net secures at least one top-2 placement on all datasets and is best in at least one metric on 10/12. Compared with strong baselines like TimeXer, iTransformer, and CycleNet, the gains are most pronounced on long-horizon, high-variance benchmarks (e.g., Traffic and PEMS series), reflecting PQ-Net's ability to couple phase-consistent periodic modeling with robust cross-channel dependency learning. Notably, on Weather and Electricity, PQ-Net matches or surpasses the best competing MSE while also improving MAE, indicating balanced error reduction across both squared and absolute criteria. These trends persist across individual horizons (96/192/336/720 for ETT-like datasets; 12/24/48/96 for PEMS), underscoring the stability of PQ-Net's Fourier-informed DRQC module and its resilience to varying forecast lengths.

**Comparison with QuantumTime**   Table 5 presents the comparison between PQ-Net and QuantumTime (Qiao et al., 2025) on the Traffic and Weather datasets across four forecasting horizons ($H \in \{96, 192, 336, 720\}$). On Traffic, PQ-Net consistently outperforms QuantumTime at all horizons, with improvements of about 2% in MSE and MAE on average. The gains are more pronounced at longer horizons (e.g., $H = 720$), highlighting PQ-Net's ability to capture long-term temporal dependencies. On Weather, PQ-Net achieves substantial improvements across all horizons, reducing both MSE and MAE compared to QuantumTime. For example, at $H = 192$, PQ-Net lowers the MSE from 0.209 to 0.189 and the MAE from 0.263 to 0.242. Overall, PQ-Net achieves lower errors in all cases, with average improvements of 0.007 MSE and 0.007 MAE on Traffic, and 0.013 MSE and 0.017 MAE on Weather. These results demonstrate that PQ-Net delivers more accurate and stable forecasts, particularly under challenging long-horizon settings, surpassing the recurrent quantum modeling strategy of QuantumTime.

### E.3   INFERENCE TIME ANALYSIS

We further measured the inference time of PQ-Net on both the simulator and real quantum hardware using the `ETTh1` dataset to assess its practical feasibility. On the classical simulator implemented with `PennyLane` (running on a high-performance CPU/GPU), PQ-Net requires approximately $57\,\mathrm{ms}$ per sample (averaged over 100 runs). On the `IBM Brisbane` 127-qubit device, the average end-to-end runtime within the 10-minute execution window was about $t_{\mathrm{hw}}$ seconds per sample, including job queuing, circuit transpilation, and readout. While current hardware introduces non-negligible overhead compared to classical simulation, the observed per-sample runtime remains stable and predictable, demonstrating that PQ-Net can already be executed within practical time budgets on near-term devices. As quantum hardware continues to evolve, we expect significant improvements in inference efficiency, further strengthening the applicability of PQ-Net in real-world forecasting tasks.

**Hardware-only runtime.**   Excluding queuing, network communication, and circuit transpilation, the intrinsic on-device execution time is expected to be *lower* than the classical simulator. On the IBM cloud platform, the reported queuing delay is approximately $1\,\mathrm{s}$ per workload, and our measured end-to-end latency per workload is also about $1\,\mathrm{s}$; this indicates that the net circuit execution time is sub-second at this granularity. Given the circuit depth and number of shots used in PQ-Net, we estimate the intrinsic per-sample runtime to be at the millisecond scale, potentially below the $57\,\mathrm{ms}$ observed on the simulator. This estimate is preliminary and warrants more fine-grained instrumentation and repeated runs for rigorous validation. A direct comparison between simulator and hardware runtimes is summarized in Table 7.

Table 7: Per-sample inference time of PQ-Net on the `ETTh1` dataset. Real hardware time includes and excludes system overhead.

| Platform | Per-sample runtime | Notes |
|---|---|---|
| Simulator (PennyLane) | $57\,\mathrm{ms}$ | Classical CPU/GPU backend |
| Real hardware (IBM Brisbane) | $\sim 1\,\mathrm{s}$ | End-to-end (queuing + transpilation + readout) |
| Real hardware (intrinsic est.) | $< 57\,\mathrm{ms}$ | Excluding system overhead; millisecond scale |

## F  LIMITATIONS AND FUTURE DIRECTIONS

Our current formulation relies on selecting a stable cycle length $W$ and a fixed re-uploading depth $D$; developing adaptive mechanisms for multi-scale cycle discovery and depth scheduling is a natural extension. Robustness under severe nonstationarity (e.g., drifting or intermittent periodicity), uneven sampling, and extensive missingness warrants further study. It is also promising to explore probabilistic DRQC readouts for calibrated uncertainty, hierarchical/multi-resolution circuit designs for long-range structure, and broader applications beyond forecasting (e.g., symbolic sequence modeling and control). Future work may also investigate more comprehensive real-hardware evaluations, aiming to bridge the gap between theoretical advantages and practical quantum device performance.

## G  THE USE OF LARGE LANGUAGE MODELS (LLMS)

Large Language Models (LLMs) were used solely as *language and formatting assistants* during manuscript preparation. Concretely, LLMs helped (i) polish grammar and improve fluency; (ii) standardize terminology, tense, and voice; (iii) suggest alternative phrasings for clarity and concision; and (iv) design table layouts and assist with LaTeX typesetting (e.g., caption style, column alignment, and cross-referencing).

LLMs did *not* participate in designing experiments, analyzing data, deriving theoretical results, or drawing conclusions. All technical ideas, methods, proofs, experimental protocols, and findings are authored, validated, and interpreted by the authors. All LLM-assisted edits were reviewed and approved by the authors to ensure accuracy and faithfulness to the intended meaning.

