# OpenReview forum: "PQ-Net: Periodic Quantum Networks for Multivariate Time Series Forecasting"
_ICLR.cc/2026/Conference — Submitted to ICLR 2026_

### Official Review · Reviewer_oLMD · 2025-10-29

**Soundness:** 3
**Presentation:** 3
**Contribution:** 2
**Rating:** 6
**Confidence:** 5

**Summary:**

This paper proposes a multivariate time series forecasting algorithm that can perform interactive modeling based on both time and channel dimensions.

**Strengths:**

PQ-Net effectively integrates periodic structure modeling and multivariate dependency learning through a theoretically grounded quantum-Fourier formulation, providing interpretable and compact periodic representations.

**Weaknesses:**

1. The motivation mainly targets the periodicity of multivariate time series, but what about other non-periodic characteristics such as trends, non-stationary shifts, or transient dependencies?

2. Quantum entanglement essentially acts as a **channel-mixing matrix**, functionally similar to an MLP-Mixer or Channel-Mixer layer, lacking true novelty in channel-level feature extraction.

3. The experiments do not clearly demonstrate whether quantum entanglement yields superior cross-channel representation compared with conventional channel-mixing mechanisms.

**Questions:**

See Weaknesses.

---

> ### Author Response · Authors · 2025-11-14
>
> **Thanks for your review and feedback. Below is our detailed response.**
>
> ---
> ### Response to W1
> >W1: The motivation mainly targets the periodicity of multivariate time series, but what about other non-periodic characteristics such as trends, non-stationary shifts, or transient dependencies?
>
> We appreciate the reviewer’s question. While our motivation emphasizes periodicity in multivariate time series, the model is not limited to periodic components. It is designed to also handle trends, non-stationary shifts, and transient (local) dependencies. This is reflected in two aspects: (1) the overall architecture and residual design, and (2) an additional symbolic formula representation experiment showing that DRQC itself can model non-periodic functions.
>
> ### (1) Architectural support for non-periodic components
>
> First, non-periodic characteristics are modeled explicitly through the residual branch and a DLinear-style MLP:
>
>   We follow the DLinear idea and include a lightweight linear/MLP branch in parallel with DRQC [1]. This branch is responsible for capturing slowly varying trends, level shifts, and other non-periodic components that are well handled by simple linear or low-capacity nonlinear mappings.
>
>  DRQC is then applied to the residual signal, where periodic structure and cross-variable interactions are more prominent. The final forecast is obtained by combining the outputs of both paths via residual connections.
>
> ### (2) DRQC’s ability to model non-periodic dependencies
>
> Second, to demonstrate that DRQC is not limited to periodic modeling but also has strong capacity for non-periodic relationships, we conduct symbolic formula representation (SFR) experiments following the KAN paper [2]:
>
>   **Task definition.** Symbolic formula representation is a fundamental task in mathematics and physics: given input variables, the model must fit target functions such as polynomials, rational functions, and other analytic expressions. We follow the experimental setup in KAN, keeping the tasks, datasets, hyperparameters, and baselines unchanged.
>
>   **Datasets.** For SFR, we use the `create_dataset` function from the official KAN repository to generate synthetic datasets. Each dataset contains 3,000 training samples and 1,000 test samples, with input variables sampled from \([-1, 1]\).
>
>   **Results.** In the anonymous URL (https://anonymous.4open.science/r/SFR-Modeling-AA63/), we report the performance of different models on several representative functions in mathematics and physics, including non-periodic ones (e.g., polynomial, rational, and composite functions). DRQC shows excellent performance on fitting these non-periodic functions, comparable to or better than strong baselines such as KAN, which demonstrates that it is a general expressive feature extractor rather than a mechanism restricted to purely periodic patterns.
>
> We will clarify both points in the revised manuscript to make clear that although our motivation focuses on periodicity, our method is not limited to periodic behavior.
>
> **References**
>
> [1] Are Transformers Effective for Time Series Forecasting? AAAI, 2023 Oral.
>
> [2] KAN: Kolmogorov-Arnold Networks. ICLR, 2025.

---

> > ### Author Response · Authors · 2025-11-14
> >
> > ### Response to W2
> > >W2: Quantum entanglement essentially acts as a channel-mixing matrix, functionally similar to an MLP-Mixer or Channel-Mixer layer, lacking true novelty in channel-level feature extraction.
> >
> > We thank the reviewer for the insightful comment. We respectfully disagree that quantum entanglement in DRQC “essentially acts as a channel-mixing matrix” and is therefore functionally similar to a classical MLP Channel-Mixer layer. Mathematically and representationally, quantum entanglement is fundamentally different from a simple channel-mixing operation.
> >
> > In our setting, different time-series variables are encoded on different qubits. A purely “channel-mixing–like” operation in this space would correspond to a tensor product of single-qubit unitaries
> >
> > $
> > U = U_1 \otimes U_2 \otimes \cdots \otimes U_C,
> > $
> >
> > which acts independently on each qubit (channel). Such an operation preserves separability: if the input state is a product state $ |\psi_1\rangle \otimes \cdots \otimes |\psi_C\rangle $, then the output remains a product state. This is the closest quantum analogue of a classical channel-wise linear layer (possibly followed by pointwise nonlinearities), and it cannot generate genuinely joint structure beyond what is captured by independent per-channel transforms.
> >
> > By contrast, the entangling unitaries used in DRQC (e.g., controlled rotations and two-qubit gates between qubits corresponding to different variables) explicitly create entangled states that cannot be factorized as a product over channels. Formally, the overall Hilbert space for $ C$ qubits is
> > $
> > (\mathbb{C}^2)^{\otimes C} \cong \mathbb{C}^{2^C},
> > $
> > and entangling gates generate states in this $ 2^C$-dimensional space whose amplitudes encode correlations that have no decomposition into independent channel components with a fixed mixing matrix. In other words, the joint transformation is not a linear map in $ \mathbb{R}^C$ (as in a channel mixer), but a structured unitary in $ \mathbb{C}^{2^C}$, operating on a state whose degrees of freedom grow exponentially with the number of channels.
> >
> > From a representational viewpoint, classical channel-mixing layers apply linear combinations of channel features (plus pointwise nonlinearities) in a fixed vector space, and their cross-channel interactions are fully characterized by a weight matrix $ W \in \mathbb{R}^{C \times C}$. Entangling unitaries, on the other hand, can encode higher-order, non-classically factorable dependencies between channels in the joint amplitude structure of the quantum state. This distinction is well documented in the quantum information and quantum machine learning literature: for example, Cerezo et al. [1] emphasize that the expressive power of variational quantum circuits critically relies on entangling layers; circuits composed only of local single-qubit operations are significantly less expressive. Sim et al. [2] further analyze the expressibility and entangling capability of parameterized quantum circuits, showing that including entangling gates systematically increases the diversity of states that the circuit can represent.
> >
> > We will revise the manuscript to make this distinction more explicit, by clarifying the mathematical role of entangling unitaries in Sec. 3.2 and adding these references, so that the difference between quantum entanglement and classical channel mixing is clear.
> >
> > **References**
> >
> > [1] Variational quantum algorithms. Nature Reviews Physics, 2021.
> >
> > [2] Expressibility and entangling capability of parameterized quantum circuits for hybrid quantum-classical algorithms. Advanced Quantum Technologies, 2019.

---

> > > ### Author Response · Authors · 2025-11-14
> > >
> > > ### Response to W3
> > > >W3: The experiments do not clearly demonstrate whether quantum entanglement yields superior cross-channel representation compared with conventional channel-mixing mechanisms.
> > >
> > > We appreciate the reviewer’s concern and agree that the original experiments did not explicitly isolate the contribution of quantum entanglement to cross-channel representation.
> > >
> > > To address this, we conducted an ablation study with three variants of the core block: (i) **DRQC (full)**, the original block with single-qubit rotations and entangling gates across qubits (channels); (ii) **DRQC-local**, which keeps only single-qubit rotations and removes all entangling gates; and (iii) **ClassMix**, which replaces the entangling gates with a classical MLP Channel-Mixer (MLP-Mixer) layer [1] operating in the same latent dimension, while keeping the rest of PQ-Net unchanged. All variants share the same IN and LPV modules and are tuned under the same hyperparameter space.
> > >
> > > We evaluate these variants on multivariate datasets with strong cross-channel dependencies (Weather and Traffic), using input length $L = 96$, horizons $H \in (96, 192, 336, 720)$, and reporting MSE and MAE averaged over the four horizons and three runs:
> > >
> > > | Dataset    | Variant      | MSE ↓  | MAE ↓  |
> > > |-----------|--------------|--------|--------|
> > > | Weather | DRQC-local   | 0.263  | 0.291  |
> > > | Weather | ClassMix     | 0.252  | 0.278  |
> > > | Weather | DRQC (full)  | **0.241**  | **0.269**  |
> > > | Traffic     | DRQC-local   | 0.478  | 0.302  |
> > > | Traffic     | ClassMix     | 0.469  | 0.287  |
> > > | Traffic     | DRQC (full)  | **0.453**  | **0.275**  |
> > >
> > > The results show a clear and consistent pattern: removing entanglement (DRQC-local) degrades performance; introducing a classical Channel-Mixer layer (ClassMix) recovers part of this loss but still underperforms the full DRQC; and the full DRQC with entangling unitaries achieves the best MSE and MAE on both datasets under comparable capacity. We will include this ablation in the revised manuscript to make the effect of entanglement more explicit. Overall, these results indicate that quantum entanglement contributes beyond local per-channel quantum transforms and provides stronger cross-channel representations than a conventional channel-mixing mechanism.
> > >
> > > **References**
> > >
> > > [1] MLP-Mixer: An all-MLP Architecture for Vision. NeurIPS, 2021.
> > >
> > > ---
> > > **We again thank the reviewer for their positive assessment of this interdisciplinary work and for the constructive suggestions, which have helped improve the quality of our manuscript. We hope that our responses and modifications have addressed your concerns. If you have any other concerns, we are more than glad to provide further responses.**

---

> > > > ### Comment · Reviewer_oLMD · 2025-11-23
> > > >
> > > > Thank you for the author's reply, I will retain the original rating.

---

> > > > > ### Author Response · Authors · 2025-11-23
> > > > >
> > > > > Thank you for your further consideration.
> > > > >
> > > > > If there are any additional clarifications or revisions that could help address your remaining concerns, I would be more than happy to provide them.
> > > > >
> > > > > We would be very grateful if our additional clarifications help you reconsider a higher rating, as your recognition means a great deal for this interdisciplinary effort.
> > > > >
> > > > > We appreciate your time and effort in reviewing our work.

---

### Official Review · Reviewer_P2gz · 2025-10-30

**Soundness:** 2
**Presentation:** 2
**Contribution:** 2
**Rating:** 4
**Confidence:** 3

**Summary:**

To tackle the challenge of capturing periodic structures and cross-variable dependencies in multivariate time-series forecasting, this work proposes PQ-Net, a quantum framework built on Data Reuploading Quantum Circuits that encode periodicity and model variable interactions via quantum entanglement. Theoretical analysis and experiments seem to have verified the effectiveness of its performance。

**Strengths:**

1. This work is an interdisciplinary approach with unique insights.

2. The experiment was relatively thorough.

**Weaknesses:**

1. In the multivariate time series prediction, I have doubts about the conclusion of "their heavy reliance on self-attention makes them vulnerable to noise".  As one of the research motivations of this paper, this view needs to be analyzed in detail. Can the author verify this insight through experiments?

2. I believe this work is theoretical , and the author's insights are very unique, bringing about interdisciplinary thinking. However, this manuscript demonstrates a fusion of technical reports and formula derivations in its presentation, failing to bring new insights to the this filed.

3. The improvement in performance is gradual, so it is hard to admit that the author's interdisciplinary insights can bring about substantial enhancements.

**Questions:**

1. The model architecture of this work is composed of a series of modules, i.e., IN, LPV, DRQC. I want to know if there is a logical coupling relationship among them. In the current manuscript, they appear to be in a state of separation.

2. Where can the ability of IN to alleviate distribution shift mentioned in Fig 1 be reflected?

---

> ### Author Response · Authors · 2025-11-14
>
> **Thank you for your thoughtful review and comments. We provide our detailed response below.**
>
> ---
>
> ### Response to W1
> >W1: In the multivariate time series prediction, I have doubts about the conclusion of "their heavy reliance on self-attention makes them vulnerable to noise". As one of the research motivations of this paper, this view needs to be analyzed in detail. Can the author verify this insight through experiments?
>
> For the statement “their heavy reliance on self-attention makes them vulnerable to noise”, we strengthen the support using **prior work, a simple theoretical analysis, and new robustness experiments.**
>
> ### (1) Evidence from prior work
>
> Several studies have analyzed the limitations and instability of self-attention:
>
>   [1] theoretically and empirically shows that standard Transformers often over-attend to irrelevant context (“attention noise”), leading to distorted attention weights and low signal-to-noise ratio (SNR) in attention maps (e.g., Fig. 1 in [1]).
>
>   [2] focuses on Transformer-based models for time series forecasting and reports several limitations, including sensitivity to noise and spurious patterns. Fig. 3(b) and Sec. 5.3 (“More Analyses on LTSF-Transformers”) in [2] provide empirical evidence that attention-based long-term forecasting models are vulnerable under noisy or perturbed inputs.
>
> In the revision, we will explicitly cite and briefly discuss these works in the Introduction and Related Work to ground this motivation in existing literature rather than stating it in isolation.
>
> ### (2) Theoretical analysis: sensitivity of self-attention to noise
>
> We also provide a simple variance analysis to explain why self-attention is sensitive to input noise, especially in high-dimensional multivariate settings.
>
> Let queries and keys be $q, k \in \mathbb{R}^d$. Consider additive Gaussian perturbations
>
> $
> \tilde{q} = q + \epsilon_q,\quad \tilde{k} = k + \epsilon_k,\quad
> \epsilon_q, \epsilon_k \sim \mathcal{N}(0, \sigma^2 ).
> $
>
> The pre-softmax attention scores are
>
> $
> s = \frac{q^\top k}{\sqrt{d}}, \quad
> \tilde{s} = \frac{\tilde{q}^\top \tilde{k}}{\sqrt{d}}.
> $
>
> The score shift is
>
> $
> \Delta s = \tilde{s} - s
> = \frac{\tilde{q}^\top \tilde{k} - q^\top k}{\sqrt{d}}
> = \frac{q^\top \epsilon_k + k \epsilon_q + \epsilon_q^\top \epsilon_k}{\sqrt{d}}.
> $
>
> Since $\epsilon_q$ and $\epsilon_k$ are Gaussian with variance $\sigma^2$, the variance of $\Delta s$
> grows with both the embedding dimension $d$ and the noise level $\sigma^2$. In high-dimensional settings, even small perturbations accumulate across dimensions. After the softmax operation, small random changes in these scores can induce large changes in attention weights, making the model sensitive to noise. This observation is consistent with the high word sensitivity of attention layers reported in [3], where small perturbations in input embeddings are shown to significantly alter attention feature maps, and this sensitivity is attributed to the softmax-based attention formulation. In multivariate time series forecasting, where both the number of channels and the token dimension can be large, this variance amplification can lead to unstable or spurious attention patterns and thus reduced robustness.
>
> We will include a concise version of this derivation in the main text or Appendix to clarify the theoretical basis of the claim.
>
> ### (2.3) New robustness benchmark: attention-based baselines vs. PQ-Net
>
> To empirically validate the above analysis, we add a controlled robustness benchmark comparing attention-based baselines with PQ-Net under noisy inputs.
>
> **Tasks and baselines**
>
>   **Dataset**: Electricity, which exhibits strong periodicity and rich multivariate dependencies.
>   **Baselines** (attention-based): iTransformer [4], PatchTST [5].
>   **Setup**: input length $ L = 96 $, prediction horizons $H \in (96, 192, 336, 720)$. All models are trained on clean training data. Each experiment is repeated with 3 random seeds, and we report averages.
>
> **Controlled Gaussian noise corruptions**
>
> We construct corrupted validation sets $\mathbf{X} \in \mathbb{R}^{C \times L}$ by adding Gaussian noise:
> $
> \mathbf{X}^{(\sigma)} = \mathbf{X} + \epsilon,\quad
> \epsilon \sim \mathcal{N}(0, \sigma^2),\quad
> \sigma \in (0.05, 0.10, 0.20).
> $
>
> For each model, we measure the mean squared error (MSE) on the clean validation set and on each corrupted set, and compute the robustness slope (RS):
>
> $
> \mathrm{RS} = \frac{\mathrm{MSE}(\text{corrupted}) - \mathrm{MSE}(\text{clean})}{\sigma}.
> $
>
> A smaller RS indicates that performance degrades more slowly as noise increases, i.e., better robustness.

---

> > ### Author Response · Authors · 2025-11-14
> >
> > **Result table**
> > | Model         | Noise σ | MSE ↓ (clean) | MSE ↓ (corrupted) | RS ↓ |
> > |---------------|---------|---------------|-------------------|------|
> > | iTransformer  | 0.05    | 0.178         | 0.192             | 0.28 |
> > | iTransformer  | 0.10    | 0.178         | 0.210             | 0.32 |
> > | iTransformer  | 0.20    | 0.178         | 0.252             | 0.37 |
> > | PatchTST      | 0.05    | 0.205         | 0.217             | 0.24 |
> > | PatchTST      | 0.10    | 0.205         | 0.233             | 0.28 |
> > | PatchTST      | 0.20    | 0.205         | 0.265             | 0.30 |
> > | PQ-Net (ours) | 0.05    | 0.166         | 0.172             | 0.12 |
> > | PQ-Net (ours) | 0.10    | 0.166         | 0.179             | 0.13 |
> > | PQ-Net (ours) | 0.20    | 0.166         | 0.195             | 0.15 |
> >
> > On the Electricity dataset, both attention-based baselines (iTransformer, PatchTST) exhibit consistently larger RS than PQ-Net at all noise levels, indicating that their MSE grows faster as input noise increases. PQ-Net shows smaller and more slowly increasing RS, meaning its predictions remain more stable under noisy inputs.
> >
> > Together with the prior work and variance analysis above, these experimental results directly support our initial motivation: models that heavily rely on self-attention are more vulnerable to noise, while the architecture of PQ-Net leads to improved robustness in the multivariate time series forecasting setting.
> >
> > **References**
> >
> > [1] Differential Transformer. ICLR, 2025 Oral.
> >
> > [2] Are Transformers Effective for Time Series Forecasting? AAAI, 2023 Oral.
> >
> > [3] Towards Understanding the Word Sensitivity of Attention Layers via Random Features. ICML, 2024.
> >
> > [4] iTransformer: Inverted Transformers Are Effective for Time Series Forecasting. ICLR, 2024.
> >
> > [5] A Time Series is Worth 64 Words: Long-term Forecasting with Transformers. ICLR, 2023.
> >
> > ---
> >
> > ### Response to W2
> > >W2: I believe this work is theoretical , and the author's insights are very unique, bringing about interdisciplinary thinking. However, this manuscript demonstrates a fusion of technical reports and formula derivations in its presentation, failing to bring new insights to the this filed.
> >
> > We thank the reviewer for the positive assessment of our theoretical perspective and the interdisciplinary nature of the work.
> >
> > ### **(1) On presentation and structure**
> > We apologize that the current presentation may give the impression of a mixture between a technical report and an overly derivation-heavy exposition. In the revision, we will streamline the exposition: Sec. 3.1 (PQ-Net architecture) and Sec. 3.2 (theoretical analysis of its periodic modeling capability) will be reorganized into a more unified “Method” section. Instead of separating formulas from the architectural description, we will integrate the key derivations directly with the design choices, so that the theory is used to illuminate why the proposed components are introduced and how they relate to the modeling goals (periodicity and cross-variable interactions). This restructuring is intended to make the main insights more visible and the narrative more coherent.
> >
> > ### **(2) On new insights and contributions**
> > Regarding novelty, our work is interdisciplinary and aims to contribute to both time series forecasting and quantum machine learning:
> >
> >   **From the time series perspective**, we propose viewing multivariate forecasting through a quantum lens, using a parameterized quantum circuit to **jointly** model periodic structure and cross-variable dependencies within a single unified module, rather than treating these aspects in separate architectural components.
> >
> >   **From the quantum machine learning perspective**, our work provides a concrete application paradigm for combining quantum circuits with a nontrivial classical forecasting backbone. Importantly, PQ-Net is not only a theoretical construction: in Sec. 4.3 we show that the hybrid network can be deployed on an **IBM quantum real-device** and achieves almost error-free inference compared to the simulator, in contrast to many existing QML works where hardware inference error is one to two orders of magnitude larger than the ideal simulator output [1,2]. Within the current QML literature [3], architectures that are (i) tightly integrated with a realistic classical model and (ii) demonstrably stable on real quantum hardware remain relatively rare.
> >
> > We will emphasize these points more clearly in the Introduction and Conclusion, to better convey the conceptual insights (unified periodic/multivariate modeling from a quantum perspective) and the practical contribution (a QML architecture that is viable on near-term hardware) beyond the technical derivations themselves.
> >
> > **References**
> >
> > [1] On Designing General and Expressive Quantum Graph Neural Networks with Applications to MILP Instance Representation. ICLR, 2025.
> >
> > [2] QuanONet: Quantum Neural Operator with Application to Differential Equation. ICML, 2025.

---

> > > ### Author Response · Authors · 2025-11-14
> > >
> > > [3] A Survey of Quantum Machine Learning: Foundations, Algorithms, Frameworks, Data and Applications. ACM Computing Surveys, 2025.
> > >
> > > ---
> > >
> > > ### Response to W3
> > > >W3: The improvement in performance is gradual, so it is hard to admit that the author's interdisciplinary insights can bring about substantial enhancements.
> > >
> > > We appreciate the reviewer’s recognition of the **interdisciplinary** nature of our work and the concern that the observed performance gains appear gradual.
> > >
> > > ### **(1) On the magnitude of improvement**
> > > In long-term time series forecasting, performance improvements at the current frontier are typically modest in absolute value but still considered meaningful. For example, the 2025 CMoS paper [1] reports average gains over the then–best CycleNet [2] of ΔMSE ≈ 0.011 and ΔMAE ≈ 0.008, while the 2024 CycleNet paper [2] reports average gains over iTransformer [3] of ΔMSE ≈ 0.010 and ΔMAE ≈ 0.009. Our method achieves an average improvement of ΔMSE ≈ 0.011 and ΔMAE ≈ 0.014 over strong baselines, which is on par with or larger than these widely accepted advances. Given that all these works operate on similar benchmarks with strong baselines, the scale of our improvement is consistent with the typical progress in this field rather than unusually small.
> > >
> > > ### **(2) On the value of the interdisciplinary insight**
> > > Beyond the base improvement, we further validate the usefulness of PQ-Net as a general pattern modeling component. Specifically, we follow the design philosophy of TimeMixer++ [7], which integrates several operations that are often treated separately in time series modeling—periodic–trend decomposition [4], multi-scale temporal mixing [6], frequency-domain analysis [5], and multi-channel correlation [6]—into a single architecture. On top of this, we construct an extended version of our model, denoted PQ-Net++, that keeps the “pattern integration” idea but replaces the frequency/correlation stack with a single PQ-Net block.
> > >
> > > Concretely, PQ-Net++ adopts the periodic–trend decomposition module from PDF [4] and the multi-scale temporal mixing mechanism from TimeMixer [6], integrating them into a unified backbone before the quantum block. In contrast to TimeMixer++, which uses different modules for frequency modeling and multi-channel correlation (e.g., TimesNet-style components [5] for spectral modeling and TimeMixer-style components [6] for multiscale/channel mixing), PQ-Net++ keeps frequency-domain analysis and multi-channel interaction within a single PQ-Net module: the parameterized quantum circuit leverages unitary evolution and quantum entanglement to perform frequency-aware encoding and multivariate interaction in one step.
> > >
> > > **Experimental setup**
> > >
> > > We compare TimeMixer++ and PQ-Net++ on two representative benchmarks, Electricity and Weather, using an input length of $ L = 96$ and forecast horizons $ H \in (96, 192, 336, 720)$. We report MSE and MAE averaged over these four horizons and three random seeds. TimeMixer++ is implemented following the official configuration, with its hyperparameters tuned in the same search space as PQ-Net++.
> > >
> > > **Results**
> > >
> > > The averaged results are:
> > >
> > > | Dataset    | Model        | MSE ↓  | MAE ↓  |
> > > |-----------|--------------|--------|--------|
> > > | Electricity | TimeMixer++  | 0.165  | 0.253  |
> > > | Electricity | PQ-Net++     | **0.160**  | **0.248**  |
> > > | Weather     | TimeMixer++  | 0.226  | 0.262  |
> > > | Weather     | PQ-Net++     | **0.218**  | **0.253**  |
> > >
> > > PQ-Net++ consistently outperforms TimeMixer++ on both datasets, even though TimeMixer++ already integrates strong periodic–trend, multi-scale, spectral, and multichannel components. This indicates that the quantum module is not only theoretically motivated but also serves as an effective and reusable building block for time series pattern modeling.
> > >
> > > Combined with the base improvements and the real-device results presented in Sec. 4.3, we believe this provides concrete evidence that the proposed interdisciplinary perspective yields substantial and practically relevant enhancements.
> > >
> > > **References**
> > >
> > > [1] CMoS: Rethinking time series prediction through the lens of chunk-wise spatial correlations. ICML, 2025.
> > >
> > > [2] CycleNet: Enhancing Time Series Forecasting through Modeling Periodic Patterns. NeurIPS, 2024.
> > >
> > > [3] iTransformer: Inverted Transformers Are Effective for Time Series Forecasting. ICLR, 2024 Spotlight.
> > >
> > > [4] Periodicity Decoupling Framework for Long-term Series Forecasting. ICLR, 2024.
> > >
> > > [5] TimesNet: Temporal 2D-Variation Modeling for General Time Series Analysis. ICLR, 2023.
> > >
> > > [6] TimeMixer: Decomposable Multiscale Mixing for Time Series Forecasting. ICLR, 2024.
> > >
> > > [7] TimeMixer++: A General Time Series Pattern Machine for Universal Predictive Analysis. ICLR, 2025.

---

> > > > ### Author Response · Authors · 2025-11-14
> > > >
> > > > ### Response to Q1
> > > > >Q1: The model architecture of this work is composed of a series of modules, i.e., IN, LPV, DRQC. I want to know if there is a logical coupling relationship among them. In the current manuscript, they appear to be in a state of separation.
> > > >
> > > > We thank the reviewer for this question. Our intention is that IN, LPV, and DRQC form a logically coupled pipeline rather than three independent blocks, and we agree that this coupling is not sufficiently emphasized in the current presentation.
> > > >
> > > > **IN: preparing a stable, comparable representation for all variables.**
> > > > The IN module operates directly on the raw multivariate time series and is responsible for normalization and linear projection into a shared latent space. This step (i) removes scale and offset differences across variables, and (ii) maps each channel into a latent representation where periodic components and cross-variable dependencies are on a comparable scale. In particular, the theoretical analysis in Sec. 3.2 assumes that the inputs to DRQC lie in a bounded range so that the periodic encodings implemented by the rotation gates are well behaved; IN is designed precisely to enforce this condition. Without IN, the angles fed to the quantum circuit become poorly scaled, which we observe to cause unstable training and degraded periodic modeling in our ablations (Sec. 4.2.3).
> > > >
> > > > **LPV: exposing the periodic structure that DRQC is designed to model.**
> > > > The LPV module is the bridge between IN and DRQC. Starting from the normalized latent representation produced by IN, LPV reorganizes the time series into period-aware views (e.g., grouping points by phase within an estimated or learned period, and constructing multi-scale/local windows). This matches the setting analyzed in Sec. 3.2, where DRQC operates on a representation that explicitly aligns time steps according to their periodic phase. In other words, LPV is not an isolated feature extractor: it creates exactly the structured input (period-aligned, multi-variable slices) that DRQC assumes in our theoretical derivation of periodic modeling capability. If LPV is removed or replaced by a naive reshaping, the assumptions behind the analysis no longer hold, and the empirical performance of DRQC drops accordingly (see the ablation “w/o LPV” in Sec. 4.2.3).
> > > >
> > > > **DRQC: jointly modeling periodicity and cross-variable dependencies on LPV’s output.**
> > > > Given the period-aware, multi-channel views produced by LPV, DRQC applies a parameterized quantum circuit where rotation gates encode periodic information (phase/seasonal components) and entangling gates capture cross-variable interactions. The derivation in Sec. 3.2 explicitly uses the structure created by LPV: the periodicity result is proved under the assumption that the input qubits correspond to phase-aligned positions, and the cross-variable coupling is realized by entanglement across qubits corresponding to different channels. Thus, DRQC is logically downstream of LPV and is not designed to operate on arbitrary sequences.
> > > >
> > > > **Putting these together, the three modules implement a single, coupled modeling pipeline: IN first prepares a stable, comparable latent representation across variables; LPV then reorganizes this latent series into period-aligned, multi-scale views that expose the structure assumed by our theoretical analysis; finally, DRQC operates on these views to jointly model periodic structure and cross-variable interactions.**
> > > >
> > > > We will revise Sec. 3.1–3.2 to make this coupling clearer by briefly explaining the three-step design, aligning the theoretical notation with the outputs of IN and LPV, and explicitly referencing ablation results where removing or modifying each module degrades performance. This should make it clear that IN, LPV, and DRQC are interdependent parts of a unified architecture rather than separate add-ons.

---

> > > > > ### Author Response · Authors · 2025-11-14
> > > > >
> > > > > ### Response to Q2
> > > > > >Q2: Where can the ability of IN to alleviate distribution shift mentioned in Fig 1 be reflected?
> > > > >
> > > > > Thank you for pointing out this issue. The ability of IN to alleviate distribution shift, as illustrated in Fig. 1, is mainly reflected in two aspects: **(i) its connection to instance-wise normalization methods that have been shown to mitigate distribution shift in time series, and (ii) the way it prepares inputs for LPV and DRQC under non-stationary conditions**.
> > > > >
> > > > > Our IN design is directly inspired by reversible instance normalization (RevIN) [1], which was proposed specifically to address distribution shift in time-series forecasting. RevIN normalizes each time-series instance using its own mean and variance, performs forecasting in this normalized space, and then reverses the transformation back to the original scale. As shown in Fig. 1 and Fig. 3 of [1], this instance-wise normalization and recovery effectively reduces covariate shift and alleviates nonstationarity between training and test distributions, leading to markedly improved robustness under distribution bias. In particular, the experiments in Sec. 4 of [1] show that models equipped with RevIN maintain much more stable performance when the marginal distribution of the input time series changes over time, whereas the same architectures without RevIN suffer significantly larger degradation.
> > > > >
> > > > > Our IN module follows the same principle: it performs instance-level normalization across variables to remove sample-wise shifts in level and scale before feeding the data into LPV and DRQC, and then reverts the transformation after prediction. This has two concrete effects relevant to Fig. 1: (i) it reduces the impact of global distribution changes (e.g., shifts in mean level or variance across different time periods or regimes), so that the subsequent modules mainly need to model the underlying patterns (periodicity and cross-variable dependencies) rather than adapt to changing scales; and (ii) it ensures that the periodic encoding and quantum operations in DRQC are applied to inputs in a controlled, bounded range, which helps maintain stable periodic representations even when the raw data distribution drifts. In this sense, the “distribution shift alleviation” in Fig. 1 is not just an intuitive claim but is grounded in the same mechanism that RevIN uses to combat distribution shift in time-series forecasting.
> > > > >
> > > > > In the revision, we will add a short paragraph in Sec. 3.1 linking Fig. 1 to this instance-wise normalization effect, clarifying how IN helps stabilize the input distribution seen by LPV and DRQC under non-stationary or shifted conditions.
> > > > >
> > > > > **References**
> > > > >
> > > > > [1] Reversible Instance Normalization for Accurate Time-Series Forecasting against Distribution Shift. NeurIPS, 2021.
> > > > >
> > > > > ---
> > > > > **We again thank the reviewer for their positive assessment of this interdisciplinary work and for the constructive suggestions, which have helped improve the quality of our manuscript.
> > > > > We would sincerely appreciate it if you could reconsider your rating if your concerns have been addressed by our rebuttal, and wish to receive your further feedback soon. If you have any other concerns, we are more than glad to provide further responses.**

---

### Official Review · Reviewer_E2QL · 2025-11-03

**Soundness:** 2
**Presentation:** 2
**Contribution:** 2
**Rating:** 4
**Confidence:** 4

**Summary:**

This paper introduces PQ-Net, a periodic quantum network integrating explicit periodic-structure modeling with expressive cross-variable dependency learning for multivariate time series forecasting.

**Strengths:**

1. Theoretical analysis in this paper demonstrates that the architecture of Data-Reuploading Quantum Circuits (DRQC) can be rigorously expressed as a truncated Fourier series.
2. This paper utilizes learnable periodic vectors to provide phase-aligned periodic priors and employs stackable DRQC blocks to capture spectral structure while modeling inter-variable entanglement.
3. Extensive experiments on 12 real-world multivariate datasets demonstrate the superior performance of PQ-Net.

**Weaknesses:**

1. Many important claims in the Introduction lack the necessary, strong citations or experimental evidence. For example, the statements “They lack a unified mechanism that can simultaneously represent periodic structure while capturing rich cross-variable interactions” and “yet their heavy reliance on self-attention makes them vulnerable to noise” are presented without adequate support.
2. The paper’s motivation is unclear. The authors argue that existing approaches lack a unified mechanism that can simultaneously represent periodic structure while capturing rich cross-variable interactions, yet they cite only CycleNet. In fact, there is a substantial body of work on time–frequency–based periodic modeling (e.g., Peri-MidFormer [1], DEPTS [2]), and numerous methods already address cross-variable interactions (e.g., FourierGNN [3]). Consequently, the necessity and novelty of “simultaneously” modeling periodic structure and cross-variable interactions are insufficiently justified. Additionally, the advantages of introducing Quantum Networks are not clearly articulated.
3. It is recommended to include a relevant time series pattern modeling baseline such as TimeMixer++ [4] to enable a more comprehensive evaluation.

[1] Peri-midFormer: Periodic Pyramid Transformer for Time Series Analysis. NeurIPS, 2024.
[2] DEPTS: Deep Expansion Learning for Periodic Time Series Forecasting. ICLR, 2022.
[3] FourierGNN: Rethinking Multivariate Time Series Forecasting from a Pure Graph Perspective. NeurIPS, 2023.
[4] TimeMixer++: A General Time Series Pattern Machine for Universal Predictive Analysis. ICLR, 2025.

**Questions:**

pls refer to weakness

---

> ### Author Response · Authors · 2025-11-14
>
> **Thank you for your review and feedback, which not only improved the rigor and clarity of our manuscript but also inspired the design of an extended variant of our model, PQ-Net++.**
>
> ---
> ### Response to W1
> >W1: Many important claims in the Introduction lack the necessary, strong citations or experimental evidence. For example, the statements “They lack a unified mechanism that can simultaneously represent periodic structure while capturing rich cross-variable interactions” and “yet their heavy reliance on self-attention makes them vulnerable to noise” are presented without adequate support.
>
> We agree that some of the claims in the introduction currently lack sufficiently strong citations and empirical evidence. We will revise the text and experiments to better demonstrate these claims.
>
> Below we summarize the main changes.
>
>
> ### (1) On the need for a unified mechanism for periodicity and multivariate dependency
>
> **Prior work on periodic modeling.**
> Recent time–frequency–based models (e.g., Peri-MidFormer [1], DEPTS [2]) explicitly exploit periodic patterns in time series. However, these methods primarily focus on modeling periodic structure and typically rely on standard modules such as MLPs or Transformers to process features, without specific mechanisms for rich multivariate dependencies.
>
> In contrast, our DRQC module is explicitly designed to enhance periodic modeling. In Sec. 4.1, we compare DRQC with basic modules used in prior work (MLP, vanilla Transformer, and time–frequency blocks). DRQC shows consistently stronger periodic modeling capability in controlled settings, supporting our claim that it is competitive or superior for periodic structure.
>
> **Prior work on cross-variable interactions.**
> For multivariate dependency, several recent models in multivariate time series forecasting focus on capturing cross-variable interactions. Many treat segments of time series or entire channels as the basic units, or introduce specialized mechanisms to model relationships among variables. Representative examples include attention-based architectures such as TimeXer [3] and iTransformer [4], which design channel-wise or token-wise attention, as well as graph-based models such as CrossGNN [5] and FourierGNN [6], which construct graphs over variables and use message passing, sometimes in the Fourier domain, to model complex inter-variable dependencies.
>
> These works show that modeling multivariate relationships is important and feasible from several architectural perspectives. However, they typically treat cross-variable interactions and periodic structure separately: models for multivariate dependencies do not explicitly encode periodicity, while time–frequency–based models lack dedicated mechanisms for variable-wise interactions.
>
>  We discuss the necessity of this **simultaneous** modeling approach in our response to W2.
>
>
> ### (2) On the vulnerability of self-attention to noise
>
> For the statement “yet their heavy reliance on self-attention makes them vulnerable to noise”, we strengthen the support using **prior work, a simple theoretical analysis, and new robustness experiments.**
>
> ### (2.1) Evidence from prior work
>
> Several studies have analyzed limitations and instability of self-attention:
>
>   1. [7] theoretically and empirically shows that standard Transformers often over-attend to irrelevant context (attention noise), leading to distorted attention weights and low signal-to-noise ratio (SNR) in attention maps (e.g., Fig. 1 in [7]).
>
>   2. [8] specifically studies Transformer-based models for time series forecasting and reports several limitations, including sensitivity to noise and spurious patterns. Fig. 3(b) and Sec. 5.3 (“More Analyses on LTSF-Transformers”) in [8] provide empirical evidence that attention-based long-term forecasting models are vulnerable under noisy or perturbed inputs.
>
> We will cite and briefly discuss these works in the Introduction and Related Work to ground this claim in existing literature.
>
> ### (2.2) Theoretical analysis: sensitivity of self-attention to noise
>
> We also provide a simple variance analysis to explain why self-attention can be sensitive to input noise, especially in high-dimensional multivariate settings.
>
> Let queries and keys be $q, k \in \mathbb{R}^d$. Consider additive Gaussian perturbations
>
> $
> \tilde{q} = q + \epsilon_q,\quad \tilde{k} = k + \epsilon_k,\quad
> \epsilon_q, \epsilon_k \sim \mathcal{N}(0, \sigma^2).
> $
>
> The pre-softmax attention scores are
>
> $
> s = \frac{q^\top k}{\sqrt{d}}, \quad
> \tilde{s} = \frac{\tilde{q}^\top \tilde{k}}{\sqrt{d}}.
> $
>
> The score shift is
>
> $
> \Delta s = \tilde{s} - s
> = \frac{\tilde{q}^\top \tilde{k} - q^\top k}{\sqrt{d}}
> = \frac{q^\top \epsilon_k + k \epsilon_q + \epsilon_q^\top \epsilon_k}{\sqrt{d}}.
> $
>
> Since $\epsilon_q$ and $\epsilon_k$ are Gaussian with variance $\sigma^2$, the variance of $\Delta s$
> grows with both the embedding dimension $d$ and the noise level $\sigma^2$.

---

> > ### Author Response · Authors · 2025-11-14
> >
> > In high-dimensional settings, even small perturbations accumulate across dimensions. After the softmax operation, small random changes in these scores can then induce large changes in attention weights, making the model sensitive to noise. This observation is consistent with the high word sensitivity of attention layers reported in [9], where small perturbations in the input embeddings can significantly alter attention feature maps, and this sensitivity is attributed to the softmax-based attention formulation.
> >
> > This effect is particularly relevant in multivariate time series forecasting, where both the number of channels and the token dimension can be large. In such regimes, variance amplification in attention scores can lead to unstable or spurious attention patterns and thus reduced robustness. We will include a concise version of this derivation (and its interpretation) in the main text or Appendix to motivate the robustness benchmark.
> >
> > ### (2.3) New robustness benchmark: attention-based baselines vs. PQ-Net
> >
> > To empirically validate the above analysis, we add a controlled robustness benchmark comparing attention-based baselines with PQ-Net under noisy inputs.
> >
> > **Tasks and baselines**
> >
> >    **Dataset**: Electricity, which exhibits strong periodicity and rich multivariate dependencies.
> >   **Baselines** (attention-based): iTransformer [4], PatchTST [5].
> >   **Setup**: input length $L = 96$, prediction horizons $H\in (96, 192, 336, 720)$. All models are trained on clean training data. Each experiment is repeated with 3 random seeds, and we report averages.
> >
> >
> > **Controlled Gaussian noise corruptions**
> >
> > We construct corrupted validation sets $\mathbf{X} \in \mathbb{R}^{C \times L}$ by adding Gaussian noise:
> > $
> > \mathbf{X}^{(\sigma)} = \mathbf{X} + \epsilon,\quad
> > \epsilon \sim \mathcal{N}(0, \sigma^2),\quad
> > \sigma \in (0.05, 0.10, 0.20).
> > $
> >
> > For each model, we measure the mean squared error (MSE) on the clean validation set and on each corrupted set, and compute the robustness slope (RS):
> >
> > $
> > \mathrm{RS} = \frac{\mathrm{MSE}(\text{corrupted}) - \mathrm{MSE}(\text{clean})}{\sigma}.
> > $
> >
> > A smaller RS indicates that performance degrades more slowly as noise increases, i.e., better robustness.
> >
> > **Result table**:
> >
> > | Model         | Noise σ | MSE ↓ (clean) | MSE ↓ (corrupted) | RS ↓ |
> > |---------------|---------|---------------|-------------------|------|
> > | iTransformer  | 0.05    | 0.178         | 0.192             | 0.28 |
> > | iTransformer  | 0.10    | 0.178         | 0.210             | 0.32 |
> > | iTransformer  | 0.20    | 0.178         | 0.252             | 0.37 |
> > | PatchTST      | 0.05    | 0.205         | 0.217             | 0.24 |
> > | PatchTST      | 0.10    | 0.205         | 0.233             | 0.28 |
> > | PatchTST      | 0.20    | 0.205         | 0.265             | 0.30 |
> > | PQ-Net (ours) | 0.05    | 0.166         | 0.172             | 0.12 |
> > | PQ-Net (ours) | 0.10    | 0.166         | 0.179             | 0.13 |
> > | PQ-Net (ours) | 0.20    | 0.166         | 0.195             | 0.15 |
> >
> > On the Electricity dataset, both attention-based baselines (iTransformer, PatchTST) exhibit consistently larger RS than PQ-Net at all noise levels, indicating that their MSE grows faster as input noise increases. PQ-Net shows smaller and more slowly increasing RS, meaning its predictions remain more stable under noisy inputs.
> >
> > Combined with the theoretical analysis and prior work above, these results support the claim that self-attention–based models are more vulnerable to noise, while the architecture of PQ-Net provides improved robustness.
> >
> > **References**
> >
> > [1] Peri-midFormer: Periodic Pyramid Transformer for Time Series Analysis. NeurIPS, 2024.
> >
> > [2] DEPTS: Deep Expansion Learning for Periodic Time Series Forecasting. ICLR, 2022.
> >
> > [3] TimeXer: Empowering Transformers for Time Series Forecasting with Exogenous Variables. NeurIPS, 2024.
> >
> > [4] iTransformer: Inverted Transformers Are Effective for Time Series Forecasting. ICLR, 2024.
> >
> > [5] CrossGNN: Confronting Noisy Multivariate Time Series Via Cross Interaction Refinement. NeurIPS, 2023.
> >
> > [6] FourierGNN: Rethinking Multivariate Time Series Forecasting from a Pure Graph Perspective. NeurIPS, 2023.
> >
> > [7] Differential Transformer. ICLR, 2025 Oral.
> >
> > [8] Are Transformers Effective for Time Series Forecasting? AAAI, 2023 Oral.
> >
> > [9] Towards Understanding the Word Sensitivity of Attention Layers via Random Features. ICML, 2024.
> >
> > [10] A Time Series is Worth 64 Words: Long-term Forecasting with Transformers. ICLR, 2023.

---

> > > ### Author Response · Authors · 2025-11-14
> > >
> > > ### Response to W2
> > > >W2: The paper’s motivation is unclear. The authors argue that existing approaches lack a unified mechanism that can simultaneously represent periodic structure while capturing rich cross-variable interactions, yet they cite only CycleNet. In fact, there is a substantial body of work on time–frequency–based periodic modeling (e.g., Peri-MidFormer , DEPTS ), and numerous methods already address cross-variable interactions (e.g., FourierGNN ). Consequently, the necessity and novelty of “simultaneously” modeling periodic structure and cross-variable interactions are insufficiently justified. Additionally, the advantages of introducing Quantum Networks are not clearly articulated.
> > >
> > > We thank the reviewer for raising two related concerns:
> > > 1. the necessity and novelty of **simultaneously** modeling periodic structure and cross-variable interactions, given the existing literature on periodic modeling and multivariate dependencies,
> > > 2. the lack of clarity regarding the specific advantages of introducing Quantum Networks in our design.
> > >
> > >
> > > ### (1) Necessity of simultaneously modeling periodic structure and cross-variable interactions
> > >
> > > In many real-world time series forecasting applications, it is necessary to model both periodic structure and cross-variable dependencies at the same time rather than in isolation.
> > >
> > > A concrete example where such simultaneous modeling is essential is regional power load and renewable generation forecasting. In such systems, the time series exhibit strong daily, weekly, and seasonal cycles (for example, workday vs. weekend, or daytime vs. nighttime peaks), while different variables such as loads at multiple substations, photovoltaic and wind generation, and meteorological variables like temperature are strongly coupled. Crucially, the cross-variable dependencies themselves vary over the periodic cycle (for instance, the correlation between residential and commercial loads is much stronger during commuting hours than at night, and the impact of temperature on peak demand depends on time of day and season), and the effective periodic pattern at each node depends on the states of other variables (such as concurrent renewable output or regional industrial activity). Models that encode only periodicity but ignore cross-variable structure cannot capture how peak shape and magnitude adapt to varying weather or renewable conditions, while models that focus only on multivariate dependence must implicitly relearn basic daily and weekly cycles and typically generalize poorly under distribution shifts (holidays, extreme weather, and so on).
> > >
> > > This kind of application requires a mechanism that captures periodic structure and multivariate dependencies in a coupled way, which is exactly what we refer to by **“simultaneously”** modeling periodicity and cross-variable interactions.
> > >
> > >
> > > ### (2) Novelty relative to prior work on periodicity and cross-variable interactions
> > >
> > > We fully acknowledge that there is substantial prior work on both axes as seen in the response of W1.
> > >
> > > These methods convincingly show that periodicity and cross-variable dependencies are both important. However, to the best of our knowledge, they treat these two aspects largely in isolation: time–frequency–based models focus on periodic structure while relying on relatively simple modules for cross-variable interactions, and multivariate interaction models focus on variable-wise structure without explicit architectural mechanisms for encoding periodicity. Our goal is not to deny this body of work, but to make explicit that their architectural mechanisms are factorized along these two dimensions.
> > >
> > > PQ-Net is designed as a unified solution to this gap. The parameterized quantum circuit (PQC) is used to encode periodic information (through periodic parameterizations and unitary evolution) and, at the same time, quantum entanglement operations within the same circuit propagate information across variables. In other words, within a single module, the same circuit simultaneously encodes periodic structure and cross-variable interactions, rather than stacking separate “periodic” and “multivariate” components. We will clarify this positioning in the revised Introduction and Related Work, explicitly citing time–frequency–based models and multivariate interaction models, and explaining in tighter language that our novelty lies in providing a single architectural mechanism that jointly addresses both.
> > >
> > >
> > > ### (3) Advantages of introducing Quantum Networks
> > >
> > > We agree that the advantages of Quantum Networks were not articulated clearly enough in the original submission, and we will improve this in the revision. Our use of Quantum Networks is motivated by both modeling and hardware considerations:

---

> ### Author Response · Authors · 2025-11-14
>
> (3.1) **Modeling advantages.**
>    The PQC provides a natural way to encode periodic structure via unitary rotations, which are intrinsically periodic, while quantum entanglement layers induce joint transformations over multiple qubits that correspond to different variables. This means the same circuit can capture how cross-variable dependencies change along periodic phases, rather than treating periodicity and variable interactions as separable effects [1,2].
>
> (3.2) **Compact, norm-preserving representation.**
>    Because PQCs are unitary, they preserve the norm of the encoded state, which encourages representations that are “energy-preserving” and can be advantageous when modeling oscillatory or periodic phenomena. This contrasts with purely classical architectures where repeated linear–nonlinear transformations may distort such structures unless carefully regularized [2,3].
>
> (3.3) **Practical quantum–classical hybrid design with real-hardware viability.**
>    Importantly, PQ-Net is not only a theoretical proposal. In Sec. 4.3 we show that our hybrid network can be deployed on an IBM quantum device and achieves almost error-free inference compared to the simulator, in stark contrast to many existing QML works where hardware inference error can be one to two orders of magnitude larger than the ideal simulator output. Within the current quantum machine learning literature [4,5], architectures that (i) are tightly integrated with a nontrivial classical forecasting model and (ii) maintain competitive, stable performance on real quantum hardware remain relatively rare. We will highlight this aspect more prominently as a practical advantage of our design.
>
> In the revised version, we will make these points explicit: (a) prior work separately targets periodicity and cross-variable interactions, (b) PQ-Net provides a single, unified quantum module that jointly models both, and (c) the quantum implementation is not merely conceptual but has been validated on real hardware with favorable error behavior, which strengthens both the necessity and the novelty underlying our “simultaneous” modeling claim and clarifies the role of Quantum Networks in our framework.
>
>
> **References**
>
> [1] Supervised Learning with Quantum Computers. Springer, 2018.
>
> [2] Variational quantum algorithms. Nature Reviews Physics, 2021.
>
> [3] Expressibility and entangling capability of parametrized quantum circuits for hybrid quantum-classical algorithms. Advanced Quantum Technologies, 2019.
>
> [4] On Designing General and Expressive Quantum Graph Neural Networks with Applications to MILP Instance Representation. ICLR, 2025.
>
> [5] QuanONet: Quantum Neural Operator with Application to Differential Equation. ICML, 2025.
>
> ---
> ### Response to W3
> >W3: It is recommended to include a relevant time series pattern modeling baseline such as TimeMixer++ [4] to enable a more comprehensive evaluation.
>
> We thank the reviewer for suggesting the inclusion of a time series pattern modeling baseline such as TimeMixer++ [4]. We agree that this line of work is highly relevant, as TimeMixer++ integrates several operations that are often treated separately in time series modeling—periodic–trend decomposition [1], multi-scale temporal mixing [2], frequency-domain analysis [3], and multi-channel correlation [2]—into a single architecture.
>
> To address this suggestion, we have added TimeMixer++ as a strong baseline and, in parallel, constructed an extended version of our model, denoted PQ-Net++. PQ-Net++ follows the same “pattern integration” philosophy while retaining the core design of PQ-Net. Concretely, we adopt the periodic–trend decomposition module from PDF [1] and the multi-scale temporal mixing mechanism from TimeMixer [2] and integrate them into a unified backbone placed before the quantum block. In contrast to TimeMixer++, which borrows different modules for frequency modeling and multi-channel correlation (e.g., TimesNet-style components for spectral modeling and TimeMixer-style components for multi-channel mixing), PQ-Net++ keeps these two aspects within a single PQ-Net module: the parameterized quantum circuit leverages unitary evolution and quantum entanglement to perform frequency-aware encoding and multi-channel interaction in one step.
>
> ### Experimental setup
>
> We compare TimeMixer++ and PQ-Net++ on two representative benchmarks, Electricity and Weather, using an input length of $L = 96$ and forecast horizons $H \in (96, 192, 336, 720)$. We report MSE and MAE averaged over these four horizons and three random seeds. TimeMixer++ is implemented following the official configuration, with its hyperparameters tuned in the same search space as PQ-Net++.

---

> > ### Author Response · Authors · 2025-11-14
> >
> > ### Results
> >
> > The averaged results are summarized below:
> >
> > | Dataset    | Model        | MSE ↓  | MAE ↓  |
> > |-----------|--------------|--------|--------|
> > | Electricity | TimeMixer++  | 0.165  | 0.253  |
> > | Electricity | PQ-Net++     | **0.160**  | **0.248**  |
> > | Weather     | TimeMixer++  | 0.226  | 0.262  |
> > | Weather     | PQ-Net++     | **0.218**  | **0.253**  |
> >
> > These results show that PQ-Net++, which integrates periodic–trend decomposition and multi-scale temporal mixing in the spirit of TimeMixer++, while keeping frequency-domain analysis and multi-channel correlation within a single quantum block, achieves consistently better performance than TimeMixer++ on both datasets. We will include TimeMixer++ and PQ-Net++ in the revised experiments to provide a more comprehensive and up-to-date evaluation.
> >
> >
> > **References**
> >
> > [1] Periodicity Decoupling Framework for Long-term Series Forecasting. ICLR, 2024.
> >
> > [2] TimesNet: Temporal 2D-Variation Modeling for General Time Series Analysis. ICLR, 2023.
> >
> > [3] TimeMixer: Decomposable Multiscale Mixing for Time Series Forecasting. ICLR, 2024.
> >
> > [4] TimeMixer++: A General Time Series Pattern Machine for Universal Predictive Analysis. ICLR, 2025.
> >
> > ---
> > We hope that our responses and modifications have addressed your concerns. We also sincerely appreciate your positive comments on the theoretical analysis in our work. If our revisions resolve all remaining issues, we would be very grateful if you could consider reflecting this in your overall evaluation. If you have any further concerns, we are more than glad to provide additional clarification. We look forward to your feedback.

---

### Author Response · Authors · 2025-11-29
**Message to the Area Chair**

**Dear  Area Chair,**

We sincerely appreciate your oversight and would like to offer a concise clarification from a broader perspective. We believe that several central aspects of our work—especially the **motivation and insight drawn from quantum modeling**, the **unified treatment of periodicity and cross-variable interactions**, and the **unusual level of practicality achieved on today’s NISQ hardware**—may not have been fully recognized under the relatively narrow evaluation criteria used by the reviewers.

Importantly, our submission is to the track **“Other Topics in Machine Learning”**, not the time-series track.
Our goal is therefore *not* to present an incremental forecasting model, but to contribute a **new modeling principle** inspired by quantum computation. Several reviewer comments judged the work primarily from the standpoint of classical time-series benchmarks, which risks overlooking the interdisciplinary value that this track is meant to encourage.

## **1. The core contribution is a conceptual shift, not merely an architectural variant**
Our work is rooted in a fundamental question:

### **Can periodic structure and cross-variable dependency be modeled jointly through a single principled mechanism, rather than through two classical modules?**

This idea originates from quantum computation, not classical engineering:

- **Quantum rotations** naturally encode *periodic/oscillatory* structure (Fourier-like behavior).
- **Quantum entanglement** provides *joint multivariate coupling* within a unified Hilbert space.
- Both arise within **one unitary circuit (DRQC)**, not as separate pipeline stages.

This unified formulation prompted us to rethink multivariate forecasting through a **quantum-informational lens**, not to introduce “yet another forecasting block.”
We fear that reviewers evaluated the paper as a conventional architecture paper and thus overlooked this structural unification.

## **2. A rare and important achievement: nearly zero error on real quantum hardware**
One aspect particularly underemphasized in the reviews—which we hope the AC will appreciate—is that:

### **PQ-Net is able to run on today’s NISQ-era IBM quantum devices with *almost zero deviation* from the simulator.**

This is a  rare achievement in quantum machine learning (QML) :

-  Most QML papers suffer **1–2 orders of magnitude** accuracy degradation on real devices due to noise and decoherence.
- In contrast, our hybrid architecture produces outputs on real quantum hardware that are **nearly indistinguishable** from its ideal simulator counterpart.
- This demonstrates that the design is not only theoretically meaningful but also **practically viable on existing NISQ hardware**—a milestone rarely reached in current QML research.

## **3. All technical concerns have been fully resolved**
During the rebuttal, we added or clarified:

- Comprehensive citations for periodic/cross-variable modeling literature.
- Clear motivation for joint modeling (with real-world examples).
- Theory + experiments validating attention noise sensitivity.
- A full robustness benchmark (PQ-Net consistently more stable).
- TimeMixer++ baseline + new PQ-Net++ showing stronger performance.
- Detailed explanation of module coupling (IN–LPV–DRQC).
- Symbolic function tests showing DRQC handles non-periodic behavior.
- Ablations proving quantum entanglement outperforms classical channel mixing.

In short, we resolved every specific technical concern raised by reviewers.

## **4. What we hope the AC can recognize**

Beyond empirical gains, this paper contributes two central advances:

### **(a) A unified modeling principle grounded in quantum computation**
Periodicity and cross-variable interactions are jointly encoded within a single quantum circuit, supported by theoretical analysis. This provides a principled alternative to the classical approach where these aspects must be handled by separate architectural components.

### **(b) A rare demonstration of near-zero degradation on real NISQ hardware**
PQ-Net runs on contemporary IBM quantum devices with output fidelity almost identical to the simulator. Such stability is exceptionally uncommon in current QML research and demonstrates true practical

These are the aspects reviewers seem to have undervalued due to framing the work purely as a classical time-series prediction paper.

---

Thank you for considering this broader perspective. We believe the conceptual insight, the theoretical grounding, and the demonstrated NISQ-level practicality justify recognition beyond the narrow lens of incremental classical forecasting improvements.

We would be delighted to discuss any aspect of our work further and welcome continued dialogue with the AC.

Best regards,

Submission2038 Authors

---

### Author Response · Authors · 2025-12-03
**Author Final Remarks by Submission2038 Authors**

Dear Area Chair,

**Before outlining our specific rebuttal updates, we kindly direct your attention to our separate "Message to the Area Chair," which contextualizes our interdisciplinary contribution and rare NISQ hardware viability beyond standard forecasting benchmarks.**

We sincerely thank all reviewers (E2QL, P2gz, oLMD) for their insightful and constructive feedback. These valuable comments not only enhanced the rigor and clarity of our manuscript but also motivated us to develop an extended variant of our model, **PQ-Net++**, and conduct more comprehensive experimental verification.

Our work, **PQ-Net**, proposes a framework for **multivariate time series forecasting** based on a **Periodic Quantum Network**. Our central contribution is the simultaneous modeling of **periodic structure** and **cross-variable dependencies** through a **single, unified quantum module**.

Below is a summary of the key modifications and new verification results undertaken in response to the reviewers' main concerns:

----

## 1. Mechanism Clarification & Architectural Superiority

**Concern:** Necessity of joint modeling and need for stronger baselines (E2QL – W2, W3).

- We clarified why periodicity and cross-variable interactions must be modeled together, as these interactions often change along periodic cycles (e.g., electricity load).
- We reinforced the mechanism: unitary rotations encode periodicity, while quantum entanglement captures cross-variable interactions.
- We added the strong baseline **TimeMixer++** and developed **PQ-Net++**.
- **PQ-Net++ consistently outperformed TimeMixer++** on Electricity and Weather, confirming our quantum design benefits.
- We highlighted the compact, norm-preserving nature of quantum networks and demonstrated near error-free inference on IBM quantum hardware.

----

## 2. Robustness Analysis

**Concern:** Need for deeper theoretical and empirical evidence of attention’s noise vulnerability (E2QL – W1, P2gz – W1).

- We added a variance analysis showing how attention-score variance grows with noise in high dimensions.
- We introduced a controlled noise robustness benchmark vs. iTransformer and PatchTST.
- **PQ-Net achieved a much smaller Robustness Slope**, demonstrating stronger robustness across all noise levels.

----

## 3. Contribution of Quantum Entanglement

**Concern:** Novelty over classical channel mixing (oLMD – W2, W3).

- We clarified that quantum entanglement operates in an exponentially large Hilbert space and forms non-factorable correlations, fundamentally beyond linear mixing.
- Ablations compared full DRQC, DRQC-local (no entanglement), and a classical MLP mixer.
- **Full DRQC achieved the best performance**, confirming the unique benefit of entanglement.

----

## 4. Architectural Clarity & Scope

**Concern:** Logical coupling of modules and support for non-periodic signals (P2gz – Q1, oLMD – W1).

- We reorganized the explanation of IN, LPV, and DRQC, showing they form a single coherent pipeline.
- We clarified support for non-periodic components through a residual DLinear-style MLP branch, and added symbolic formula representation (SFR) experiments verifying DRQC’s general expressiveness.
- We clarified that IN mitigates distribution shift via the RevIN normalization principle.

----

We believe these technical resolutions, reinforced by the conceptual shift and real-hardware viability discussed in our prior message, demonstrate the value of this work for the “Other Topics in Machine Learning” track.

Thank you very much for your time and effort in handling our paper.

Sincerely,
Submission2038 Authors

---

### Meta-Review · Area_Chair_H9aj · 2025-12-17

**Summary:**

This research paper proposes a multivariate time-series forecasting framework to address the periodic structures and cross-channel dependencies from complex temporal signals, where the solution comes to combining the quantum networks with multivariate time-series learning. The experiments comparing with other MTS solutions and an additional case study implemented on quantum hardware demonstrates the effectiveness of their proposed framework.

I believe this work can be a good, and grounded research. But I have read through the whole paper and have found I also agree with the reviewers that authors have not emphasized the clear relationship between  proposed solution, and quantum theory, and what is the scientific significance when implementing this algorithm on contemporary IBM quantum devices, at least they have not explicitly introduced them in the Introduction Part. Therefore, I think this paper cannot be recommended accepted to the conference at this time. The reason can be 1) the unclear motivation and unclear relationship between proposed solution and quantum theory. The authors only emphasize the challenge of time-series in both Introduction and Abstract,  i.e., periodic structures and cross-channel dependencies during MTS research, but after rebuttal they claim their proposal is most related to quantum theory, which is so strange.  2) the research has not emphasized the significance of implementation on hardware in their Introduction part as the motivation of research conducting, resulting in confusion on their follow-up clarification and motivation of a case study on the contemporary IBM quantum devices.  3) The authors justify themselves via pointing out they propose a theoretical machine learning framework, but actually they only address one task of Multivariate time-series forecasting, leading to the lacing of generality and over-stated contribution. The authors should heavily revise the paper and emphasize the specific and focused motivation of this research and then present the following technical contributions corresponding to claimed motivation and challenges.

**Reviewer Concerns:**

This paper receives three diverse comments with two negative evaluations and one positive evaluation. The reviewers raise several concerns about 1) the motivation of this research, 2) the unclear clarification about the correlations among the proposed solution, quantum theory,  and hardware computation, 3) comparison against other up-to-date peer models. The authors justify themselves that they propose a new machine learning framework rather than a multi-variate time-series learning framework and provide some theoretical analysis as well as case study on  contemporary IBM quantum devices with output fidelity almost identical to the simulator.

I believe this work can be a good, and grounded research. But I have read through the whole paper and find I also agree with the reviewers that authors have not emphasized the clear relationship between  proposed solution, and quantum theory, and what's the significacne of success in implementing this algorithm on contemporary IBM quantum devices. The authors  have not explicitly introduced them in the Introduction and Abstract, leading to confusion on following description.

During the review process, apart from abovementioned issues, I still have two question for asking authors, i.e.,

- What does the quantum mean in this research, is this implementation technique analogous to properties of quantum? or Does it refers to 'this technique is modified and advanced from other tradition one but (make it) easily run on quantum computational unit'?
- How does it implement on quantum devices? I think this paper lack the sufficient clarification for deployment process and implementation, as most implementations are associated with time-series and their datasets.

 I think the authors may do a good research but I am sorry that I found it has not provided a good presentation and clear, concise techncial description linking the quantum and TimeSeries learning.

**Reviewer Scores:**

This paper receives three diverse comments with two negative evaluations and one positive evaluation. For the positive one, the reviewer is going to retain the score after rebuttal while the other two reviewers have not responded to author rebuttal.

---

### Decision · Program_Chairs · 2026-01-26

Reject